# EXPLAINING SCALING LAWS OF NEURAL NETWORK GENERALIZATION

## ABSTRACT

The test loss of well-trained neural networks often follows precise power-law scaling relations with either the size of the training dataset or the number of parameters in the network. We propose a theory that explains and connects these scaling laws. We identify *variance-limited* and *resolution-limited* scaling behavior for both dataset and model size, for a total of four scaling regimes. The variance-limited scaling follows simply from the existence of a well-behaved infinite data or infinite width limit, while the resolution-limited regime can be explained by positing that models are effectively resolving a smooth data manifold. In the large width limit, this can be equivalently obtained from the spectrum of certain kernels, and we present evidence that large width and large dataset resolution-limited scaling exponents are related by a duality. We exhibit all four scaling regimes in the controlled setting of large random feature and pretrained models and test the predictions empirically on a range of standard architectures and datasets. We also observe several empirical relationships between datasets and scaling exponents: super-classing image tasks does not change exponents, while changing input distribution (via changing datasets or adding noise) has a strong effect. We further explore the effect of architecture aspect ratio on scaling exponents.

## 1 SCALING LAWS FOR NEURAL NETWORKS

For a large variety of models and datasets, neural network performance has been empirically observed to scale as a power-law with model size and dataset size (Hestness et al., 2017; Kaplan et al., 2020; Rosenfeld et al., 2020b; Henighan et al., 2020). These exponents determine how quickly performance improves with more data and larger models. We would like to understand why these power-laws emerge, and what features of the data and models determine the values of the power law exponents.

In this work, we present a theoretical framework for understanding scaling laws in trained neural networks. We identify four related scaling regimes with respect to the number of model parameters $P$ and the dataset size $D$. With respect to each of $D, P$, there is both a *variance-limited* regime and a *resolution-limited* regime.

**Variance-Limited Regime** In the limit of infinite data or an arbitrarily wide model, some aspects of neural network training simplify. Specifically, if we fix one of $D, P$ and study scaling with respect to the other parameter as it becomes arbitrarily large, then the difference between the finite test loss and its limiting value scales as $1/x$, i.e. as a power-law with exponent 1, with $x = D$ or $\sqrt{P} \propto$ width in deep networks and $x = D$ or $P$ in linear models.

**Resolution-Limited Regime** In this regime, one of $D$ or $P$ is effectively infinite, and we study scaling as the *other* parameter increases. In this case, a variety of works have empirically observed power-law scalings $1/x^{\alpha}$, typically with $0 < \alpha < 1$ for both $x = P$ or $D$. We derive exponents in this regime precisely in the setting of random feature models (c.f. next section). Empirically, we find that our theoretical predictions for exponents hold in pretrained, fine-tuned models even though these lie outside our theoretical setting.

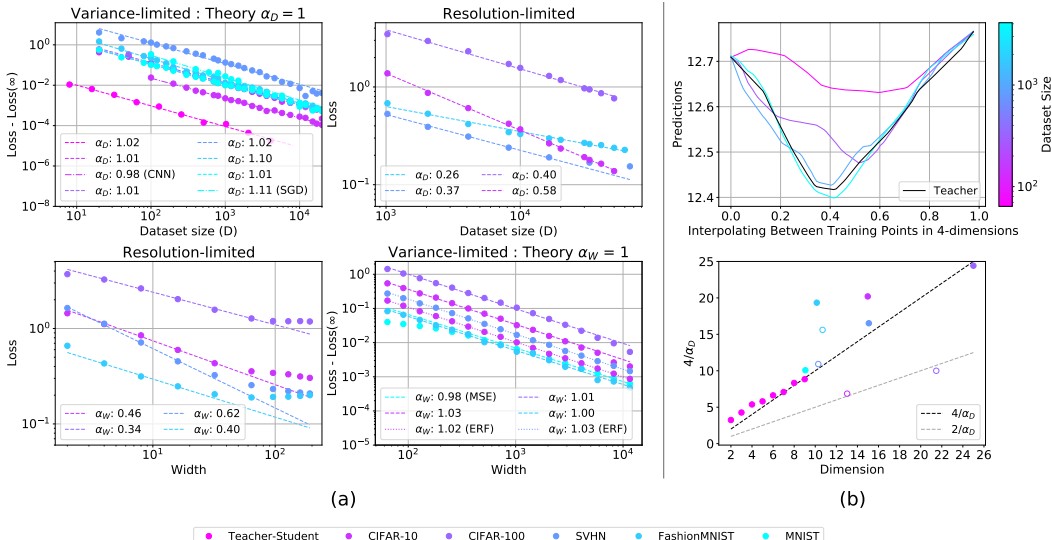

Figure 1: **(a) Four scaling regimes** Here we exhibit the four regimes we focus on in this work. **(top-left, bottom-right)** *Variance-limited* scaling of under-parameterized models with dataset size and over-parameterized models with number of parameters (width) exhibit universal scaling ($\alpha_D = \alpha_W = 1$) independent of the architecture or underlying dataset. **(top-right, bottom-left)** *Resolution-limited* over-parameterized models with dataset or under-parameterized models with model size exhibit scaling with exponents that depend on the details of the data distribution. These four regimes are also found in random feature (Figure 2a) and pretrained models (see supplement). **(b) Resolution-limited models interpolate the data manifold** Linear interpolation between two training points in a four-dimensional input space **(top)**. We show a teacher model and four student models, each trained on different sized datasets. In all cases teacher and student approximately agree on the training endpoints, but as the training set size increases they increasingly match everywhere. **(bottom)** We show $4/\alpha_D$ versus the data manifold dimension (input dimension for teacher-student models, intrinsic dimension for standard datasets). We find that the teacher-student models follow the $4/\alpha_D$ (dark dashed line), while the relationship for a four layer CNN (solid) and WRN (hollow) on standard datasets is less clear.

For more general nonlinear models, we propose a refinement of naive bounds into estimates via expansions that hold asymptotically. These rely on the idea that additional data (in the infinite model-size limit) or added model parameters (in the infinite data limit) are used by the model to carve up the data manifold into smaller components. For smooth manifolds, loss, and network, the test loss will depend on the linear size of a sub-region, while it is the $d$-dimensional sub-region volume that scales inversely with $P$ or $D$, giving rise to $\alpha \propto 1/d$.[1] To test this empirically, we make measurements of the resolution-limited exponents in neural networks and intrinsic dimension of the data manifold, shown in Figure 1b.

**Explicit Derivation** We derive the scaling laws for these four regimes explicitly in the setting of random feature teacher-student models, which also applies to neural networks in the large width limit. This setting allows us to solve for the test error directly in terms of the feature covariance (kernel). The scaling of the test loss then follows from the asymptotic decay of the spectrum of the covariance matrix. For generic continuous kernels on a $d$-dimensional manifold, we can further relate this to the dimension of the data manifold.

**Summary of Contributions:**

1. We propose four scaling regimes for neural networks. The variance-limited and resolution-limited regimes originate from different mechanisms, which we identify. To our knowledge,

---

[1] A visualization of this successively better approximation with dataset size is shown in Figure 1b for models trained to predict data generated by a random fully-connected network.

this categorization has not been previously exhibited. We provide empirical support for all four regimes in deep networks on standard datasets.

2. We derive the variance-limited regime under simple yet general assumptions (Theorem 1).

3. We present a hypothesis for resolution-limited scaling through refinement of naive bounds (Theorems 2, 3), for general nonlinear models. We empirically test the dependence of the estimates on intrinsic dimension of the data manifold for deep networks on standard datasets (Figure 1b).

4. In the setting of random feature teacher-student networks, we derive *both* variance-limited and resolution-limited scaling exponents exactly. In the latter case, we relate this to the spectral decay of kernels. We identify a novel *duality* that exists between model and dataset size scaling.

5. We empirically investigate predictions from the random features setting in pretrained, fine-tuned models on standard datasets and find they give excellent agreement.

6. We study the dependence of the scaling exponent on changes in architecture and data, finding that (i) changing the input distribution via switching datasets and (ii) the addition of noise have strong effects on the exponent, while (iii) changing the target task via superclassing does not.

**Related Works**: There have been a number of recent works demonstrating empirical scaling laws (Hestness et al., 2017; Kaplan et al., 2020; Rosenfeld et al., 2020b; Henighan et al., 2020; Rosenfeld et al., 2020a) in deep neural networks, including scaling laws with model size, dataset size, compute, and other observables such as mutual information and pruning. Some precursors (Ahmad & Tesauro, 1989; Cohn & Tesauro, 1991) can be found in earlier literature. Recently, scaling laws have also played a significant role in motivating work on the largest models that have yet been developed (Brown et al., 2020; Fedus et al., 2021).

There has been comparatively little work on theoretical ideas (Sharma & Kaplan, 2020; Bisla et al., 2021) that match and explain empirical findings in generic deep neural networks. In the particular case of large width, deep neural networks behave as random feature models (Neal, 1994; Lee et al., 2018; Matthews et al., 2018; Jacot et al., 2018; Lee et al., 2019; Dyer & Gur-Ari, 2020), and known results on the loss scaling of kernel methods can be applied (Spigler et al., 2020; Bordelon et al., 2020). Though not in the original, Bordelon et al. (2020) analyze resolution-limited dataset size scaling for power-law spectra in later versions.

During the completion of this work, Hutter (2021) presented a specific solvable model of learning exhibiting non-trivial power-law scaling for power-law (Zipf) distributed features. This does not directly relate to the setups studied in this work, or present bounds that supersede our results. Concurrent to our work, Bisla et al. (2021) presented a derivation of the resolution-limited scaling with dataset size, also stemming from nearest neighbor distance scaling on data manifolds. However, they do not discuss requirements on model versus dataset size or how this scaling behavior fits into other asymptotic scaling regimes.

In the variance-limited regime, scaling laws in the context of random feature models (Rahimi & Recht, 2008; Hastie et al., 2019; d'Ascoli et al., 2020), deep linear models (Advani & Saxe, 2017; Advani et al., 2020), one-hidden-layer networks (Mei & Montanari, 2019; Adlam & Pennington, 2020a;b), and wide neural networks treated as Gaussian processes or trained in the NTK regime (Lee et al., 2019; Dyer & Gur-Ari, 2020; Andreassen & Dyer, 2020; Geiger et al., 2020) have been studied. In particular, this behavior was used in (Kaplan et al., 2020) to motivate a particular ansatz for simultaneous scaling with data and model size. The resolution-limited analysis can perhaps be viewed as an attempt to quantify the *ideal-world* generalization error of Nakkiran et al. (2021).

This work makes use of classic results connecting the spectrum of a smooth kernel to the geometry it is defined over (Weyl, 1912; Reade, 1983; Kühn, 1987; Ferreira & Menegatto, 2009) and on the scaling of iteratively refined approximations to smooth manifolds (Stein, 1999; Bickel et al., 2007; de Laat, 2011).

## 2 FOUR SCALING REGIMES

Throughout this work we will be interested in how the average test loss $L(D, P)$ depends on the dataset size $D$ and the number of model parameters $P$. Unless otherwise noted, $L$ denotes the test loss averaged over initialization of the parameters and draws of a size $D$ training set. Some of our results only pertain directly to the scaling with width $w \propto \sqrt{P}$, but we expect many of the intuitions apply more generally. We use the notation $\alpha_D$, $\alpha_P$, and $\alpha_W$ to indicate scaling exponents with respect to dataset size, parameter count, and width. All proofs appear in the supplement.

### 2.1 VARIANCE-LIMITED EXPONENTS

In the limit of large $D$ the outputs of an appropriately trained network approach a limiting form with corrections which scale as $D^{-1}$. Similarly, recent work shows that wide networks have a smooth large $P$ limit (Jacot et al., 2018), where fluctuations scale as $1/\sqrt{P}$. If the loss is sufficiently smooth then its value will approach the asymptotic loss with corrections proportional to the variance, ($1/D$ or $1/\sqrt{P}$). In Theorem 1 we present sufficient conditions on the loss to ensure this variance dominated scaling. We note, these conditions are satisfied by mean squared error and cross entropy loss, though we conjecture the result holds even more generally.

**Theorem 1.** *Let $\ell(f)$ be the test loss as a function of network output, ($L = \mathbb{E}[\ell(f)]$), and let $f_T$ be the network output after $T$ training steps, thought of as a random variable over weight initialization, draws of the training dataset, and optimization seed. Further let $f_T$ be concentrating with $\mathbb{E}[(f_T - \mathbb{E}[f_T])^k] = \mathcal{O}(\epsilon) \, \forall k \geq 2$. If $\ell$ is a finite degree polynomial, or has bounded second derivative, or is 2-Hölder, then $\mathbb{E}[\ell(f_T)] - \ell(\mathbb{E}[f_T]) = \mathcal{O}(\epsilon)$.*

**Dataset scaling** Consider a neural network, and its associated training loss $L_{\text{train}}(\theta)$. For every value of the weights, the training loss, thought of as a random variable over draws of a training set of size $D$, concentrates around the population loss, with a variance which scales as $\mathcal{O}(D^{-1})$. If the optimization procedure is sufficiently smooth, the trained weights, network output, and higher moments, will approach their infinite $D$ values, $\mathbb{E}_D\left[(f_T - \mathbb{E}_D[f_T])^k\right] = \mathcal{O}(D^{-1})$. Here, the subscript $D$ on the expectation indicates an average over draws of the training set. This scaling together with Theorem 1 gives the variance limited scaling of loss with dataset size.

This concentration result with respect to dataset size has appeared for linear models in Rahimi & Recht (2008) and for single hidden layer networks with high-dimensional input data in Mei & Montanari (2019); Adlam & Pennington (2020a;b). In the supplement we prove this for GD and SGD with polynomial loss as well as present informal arguments more generally. Additionally, we present examples violating the smoothness assumption and exhibiting different scaling.

**Large Width Scaling** We can make a very similar argument in the $w \to \infty$ limit. It has been shown that the predictions from an infinitely wide network, either under Bayesian inference (Neal, 1994; Lee et al., 2018), or when trained via gradient descent (Jacot et al., 2018; Lee et al., 2019) approach a limiting distribution at large width equivalent to a linear model. Furthermore, corrections to the infinite width behavior are controlled by the variance of the full model around the linear model predictions. This variance (and higher moments) have been shown to scale as $1/w$ (Dyer & Gur-Ari, 2020; Yaida, 2020; Andreassen & Dyer, 2020), $\mathbb{E}_w\left[(f_T - \mathbb{E}_w[f_T])^k\right] = \mathcal{O}(w^{-1})$. Theorem 1 then implies the loss will differ from its $w = \infty$ limit by a term proportional to $1/w$.

We note that there has also been work studying the combined large depth and large width limit, where Hanin & Nica (2020) found a well-defined infinite size limit with controlled fluctuations. In any such context where the model predictions concentrate, we expect the loss to scale with the variance of the model output. In the case of linear models, studied below, the variance is $\mathcal{O}(P^{-1})$ rather than $\mathcal{O}(\sqrt{P})$, and we see the associated variance scaling in this case.

## 2.2 RESOLUTION-LIMITED EXPONENTS

In this section we consider training and test data drawn uniformly from a compact $d$-dimensional manifold, $x \in \mathcal{M}_d$, and targets given by some smooth function $y = \mathcal{F}(x)$ on this manifold.

**Over-Parameterized Dataset Scaling**

Consider the double limit of an over-parameterized model with large training set size, $P \gg D \gg 1$. We further consider *well-trained* models, i.e. models that interpolate all training data. The goal is to understand $L(D)$. If we assume that the learned model $f$ is sufficiently smooth, then the dependence of the loss on $D$ can be bounded in terms of the dimension of the data manifold $\mathcal{M}_d$.

Informally, if our train and test data are drawn i.i.d. from the same manifold, then the distance from a test point to the closest training data point decreases as we add more and more training data points. In particular, this distance scales as $\mathcal{O}(D^{-1/d})$ (Levina & Bickel, 2005). Furthermore, if $f$, $\mathcal{F}$ are both sufficiently smooth, they cannot differ too much over this distance. If in addition the loss function, $L$, is a smooth function vanishing when $f = \mathcal{F}$, we have $L = \mathcal{O}(D^{-1/d})$. This is summarized in the following theorem.

**Theorem 2.** *Let $L(f)$, $f$ and $\mathcal{F}$ be Lipschitz with constants $K_L$, $K_f$, and $K_\mathcal{F}$. Further let $\mathcal{D}$ be a training dataset of size $D$ sampled i.i.d from $\mathcal{M}_d$ and let $f(x) = \mathcal{F}(x)$, $\forall x \in \mathcal{D}$ then $L(D) = \mathcal{O}\left(K_L max(K_f, K_\mathcal{F})D^{-1/d}\right)$.*

**Under-Parameterized Parameter Scaling**

We will again assume that $\mathcal{F}$ varies smoothly on an underlying compact $d$-dimensional manifold $\mathcal{M}_d$. We can obtain a bound on $L(P)$ if we imagine that $f$ approximates $\mathcal{F}$ as a piecewise function with roughly $P$ regions (see Sharma & Kaplan (2020)). Here, we instead make use of the argument from the over-parameterized, resolution-limited regime above. If we construct a sufficiently smooth estimator for $\mathcal{F}$ by interpolating among $P$ randomly chosen points from the (arbitrarily large) training set, then by the argument above the loss will be bounded by $\mathcal{O}(P^{-1/d})$.

**Theorem 3.** *Let $L(f)$, $f$ and $\mathcal{F}$ be Lipschitz with constants $K_L$, $K_f$, and $K_\mathcal{F}$. Further let $f(x) = \mathcal{F}(x)$ for $P$ points sampled i.i.d from $\mathcal{M}_d$ then $L(P) = \mathcal{O}\left(K_L max(K_f, K_\mathcal{F})P^{-1/d}\right)$.*

**From Bounds to Estimates**

Theorems 2 and 3 are phrased as bounds, but we expect the stronger statement that these bounds also generically serve as estimates, so that eg $L(D) = \Omega(D^{-c/d})$ for $c \geq 2$, and similarly for parameter scaling. If we assume that $\mathcal{F}$ and $f$ are analytic functions on $\mathcal{M}_d$ and that the loss function $L(f, \mathcal{F})$ is analytic in $f - \mathcal{F}$ and minimized at $f = \mathcal{F}$, then the loss at a given test input, $x_{\text{test}}$, can be expanded around the nearest training point, $\hat{x}_{\text{train}}$, $L(x_{\text{test}}) = \sum_{m=n \geq 2}^{\infty} a_m(\hat{x}_{\text{train}})(x_{\text{test}} - \hat{x}_{\text{train}})^m$,[2] where the first term is of finite order $n \geq 2$ because the loss vanishes at the training point. As the typical distance between nearest neighbor points scales as $D^{-1/d}$ on a $d$-dimensional manifold, the loss will be dominated by the leading term, $L \propto D^{-n/d}$, at large $D$. Note that if the model provides an accurate piecewise linear approximation, we will generically find $n \geq 4$.

## 2.3 EXPLICIT REALIZATION IN LINEAR MODELS

In the proceeding sections we have conjectured typical case scaling relations for a model's test loss. We have further given intuitive arguments for this behavior which relied on smoothness assumptions on the loss and training procedure. In this section, we provide a concrete realization of all four scaling regimes within the context of linear models. Of particular interest is the resolution-limited regime, where the scaling of the loss is a consequence of the linear model kernel spectrum – the scaling of over-parameterized models with dataset size and under-parameterized models with parameters is a consequence of a classic result, originally due to Weyl (1912), bounding the spectrum of sufficiently smooth kernel functions by the dimension of the manifold they act on.

---

[2]For simplicity we have used a very compressed notation for multi-tensor contractions in higher order terms.

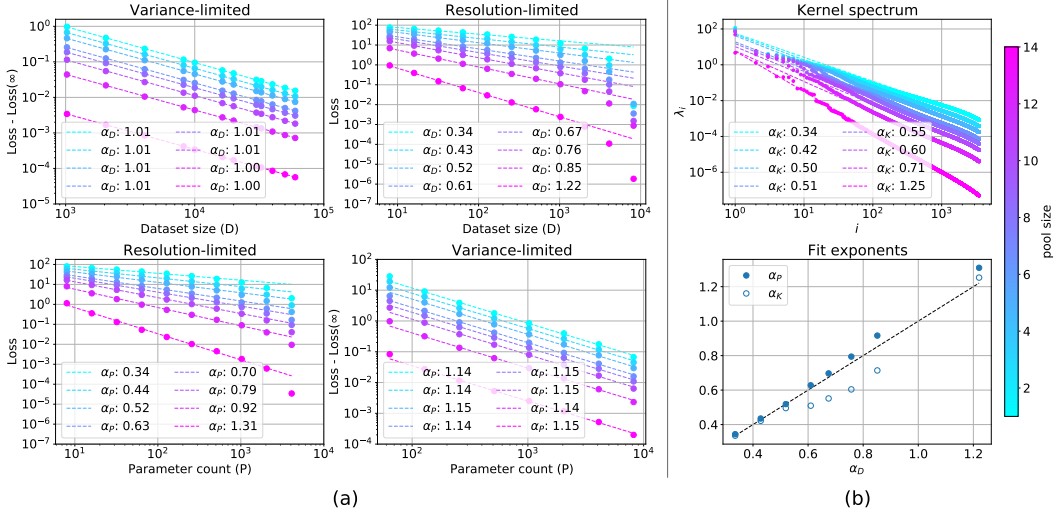

Figure 2: **(a) Random feature models exhibit all four scaling regimes** Here we consider linear teacher-student models with random features trained with MSE loss to convergence. We see both *variance-limited* scaling (**top-left, bottom-right**) and *resolution-limited* scaling (**top-right, bottom-left**). Data is varied by downsampling MNIST by the specified pool size. **(b) Duality and spectra in random feature models** Here we show the relation between the decay of the kernel spectra, $\alpha_K$, and the scaling of the loss with number of data points, $\alpha_D$, and with number of parameters, $\alpha_P$ **(top)** The spectra of random FC kernels on pooled MNIST **(bottom)** appear well described by a power law decay. The theoretical relation $\alpha_D = \alpha_P = \alpha_K$ is given by the black dashed line.

Linear predictors serve as a model system for learning. Such models are used frequently in practice when more expressive models are unnecessary or infeasible (McCullagh & Nelder, 1989; Rifkin & Lippert, 2007; Hastie et al., 2009) and also serve as an instructive test bed to study training dynamics (Advani et al., 2020; Goh, 2017; Hastie et al., 2019; Nakkiran, 2019; Grosse, 2021). Furthermore, in the large width limit, randomly initialized neural networks become Gaussian Processes (Neal, 1994; Lee et al., 2018; Matthews et al., 2018; Novak et al., 2019; Garriga-Alonso et al., 2019; Yang, 2019), and in the low-learning rate regime (Lee et al., 2019; Lewkowycz et al., 2020; Huang et al., 2020) neural networks train as linear models at infinite width (Jacot et al., 2018; Lee et al., 2019; Chizat et al., 2019).

Here we discuss linear models in general terms, though the results immediately hold for the special cases of wide neural networks. In this section we focus on teacher-student models with weights initialized to zero and trained with mean squared error (MSE) loss to their global optimum.

We consider a linear teacher, $F$, and student $f$, $F(x) = \sum_{M=1}^{S} \omega_M F_M(x)$, $f(x) = \sum_{\mu=1}^{P} \theta_\mu f_\mu(x)$. Here $\{F_M\}$ are a (potentially infinite) pool of features and the teacher weights, $\omega_M$ are taken to be normal distributed, $\omega \sim \mathcal{N}(0, 1/S)$. The student model is built out of a subset of the teacher features. To vary the number of parameters in this simple model, we construct $P$ features, $f_{\mu=1,\dots,P}$, by introducing a projector $\mathcal{P}$ onto a $P$-dimensional subspace of the teacher features, $f_\mu = \sum_M \mathcal{P}_{\mu M} F_M$.

We train by sampling a training set of size $D$ and minimizing the MSE loss, $L_{\text{train}} = \frac{1}{2D} \sum_{a=1}^{D} (f(x_a) - F(x_a))^2$. We are interested in the test loss averaged over draws of our teacher and training dataset. The infinite data test loss, $L(P) := \lim_{D \to \infty} L(D, P)$, takes the form.

$$L(P) = \frac{1}{2S} \text{Tr} \left[ \mathcal{C} - \mathcal{C}\mathcal{P}^T \left( \mathcal{P}\mathcal{C}\mathcal{P}^T \right)^{-1} \mathcal{P}\mathcal{C} \right] . \tag{1}$$

Here we have introduced the feature-feature second moment-matrix, $\mathcal{C} = \mathbb{E}_x \left[ F(x)F^T(x) \right]$.

If the teacher and student features had the same span, this would vanish, but due to the mismatch the loss is non-zero. On the other hand, if we keep a finite number of training points, but allow the

student to use all of the teacher features, the test loss, $L(D) := \lim_{P \to S} L(D, P)$, takes the form,

$$L(D) = \frac{1}{2} \mathbb{E}_x \left[ \mathcal{K}(x, x) - \vec{\mathcal{K}}(x) \bar{\mathcal{K}}^{-1} \vec{\mathcal{K}}(x) \right] . \tag{2}$$

Here, $\mathcal{K}(x, x')$ is the data-data second moment matrix, $\vec{\mathcal{K}}$ indicates restricting one argument to the $D$ training points, while $\bar{\mathcal{K}}$ indicates restricting both. This test loss vanishes as the number of training points becomes infinite but is non-zero for finite training size.

We present a full derivation of these expressions in the supplement. In the remainder of this section, we explore the scaling of the test loss with dataset and model size.

### 2.3.1 VARIANCE-LIMITED EXPONENTS

To derive the limiting expressions (1) and (2) for the loss one makes use of the fact that the sample expectation of the second moment matrix over the finite dataset, and finite feature set is close to the full covariance, $\frac{1}{D} \sum_{a=1}^{D} F(x_a) F^T(x_a) = \mathcal{C} + \delta\mathcal{C}$, $\frac{1}{P} f^T(x) f(x'), = \mathcal{K} + \delta\mathcal{K}$, with the fluctuations satisfying $\mathbb{E}_D \left[ \delta C^2 \right] = \mathcal{O}(D^{-1})$ and $\mathbb{E}_P \left[ \delta K^2 \right] = \mathcal{O}(P^{-1})$, where expectations are taken over draws of a dataset of size $D$ and over feature sets. Using these expansions yields the variance-limited scaling, $L(D, P) - L(P) = \mathcal{O}(D^{-1})$, $L(D, P) - L(D) = \mathcal{O}(P^{-1})$ in the under-parameterized and over-parameterized settings respectively.

In Figure 2a we see evidence of these scaling relations for features built from randomly initialized ReLU networks on pooled MNIST independent of the pool size. In the supplement we provide an in-depth derivation of this behavior and expressions for the leading contributions to $L(D, P) - L(P)$ and $L(D, P) - L(D)$.

### 2.3.2 RESOLUTION-LIMITED EXPONENTS

We now would like to analyze the scaling behavior of our linear model in the resolution-limited regimes, that is the scaling with $P$ when $1 \ll P \ll D$ and the scaling with $D$ when $1 \ll D \ll P$. In these cases, the scaling is controlled by the shared spectrum of $\mathcal{C}$ or $\mathcal{K}$. This spectrum is often well described by a power-law, where eigenvalues $\lambda_i$ satisfy $\lambda_i = \frac{1}{i^{1+\alpha_K}}$. See Figure 2b for example spectra on pooled MNIST. In this case, we will argue that the losses also obey a power law scaling, with the exponents controlled by the spectral decay factor, $1 + \alpha_K$.

$$L(D) \propto D^{-\alpha_K} , \quad L(P) \propto P^{-\alpha_K} . \tag{3}$$

In other words, in this setting, $\alpha_P = \alpha_D = \alpha_K$. This is supported empirically in Figure 2b. We then argue that when the kernel function, $\mathcal{K}$ is sufficiently smooth on a manifold of dimension $d$, $\alpha_K \propto d^{-1}$, thus realizing the more general resolution-limited picture described above.

**From spectra to scaling laws for the loss** To be concrete let us focus on the over-parameterized loss. If we introduce the notation $e_i$ for the eigenvectors of $\mathcal{C}$ and $\bar{e}_i$ for the eignvectors of $\frac{1}{D} \sum_{a=1}^{D} F(x_a) F^T(x_a)$, the loss becomes,

$$L(D) = \frac{1}{2} \sum_{i=1}^{S} \lambda_i (1 - \sum_{j=1}^{D} (e_i \cdot \bar{e}_j)^2) . \tag{4}$$

Before discussing the general asymptotic behavior of (4), we can gain some intuition by considering the case of large $\alpha_K$. In this case, $\bar{e}_j \approx e_j$ (see e.g. Loukas (2017)), we can simplify (4) to,

$$L(D) \propto \sum_{D+1}^{\infty} \frac{1}{i^{1+\alpha_K}} = \alpha_K D^{-\alpha_K} + \mathcal{O}(D^{-\alpha_K - 1}) . \tag{5}$$

More generally in the supplement, following Bordelon et al. (2020); Canatar et al. (2021), we use replica theory methods to derive, $L(D) \propto D^{-\alpha_K}$ and $L(P) \propto P^{-\alpha_K}$, without requiring the large $\alpha_K$ limit.

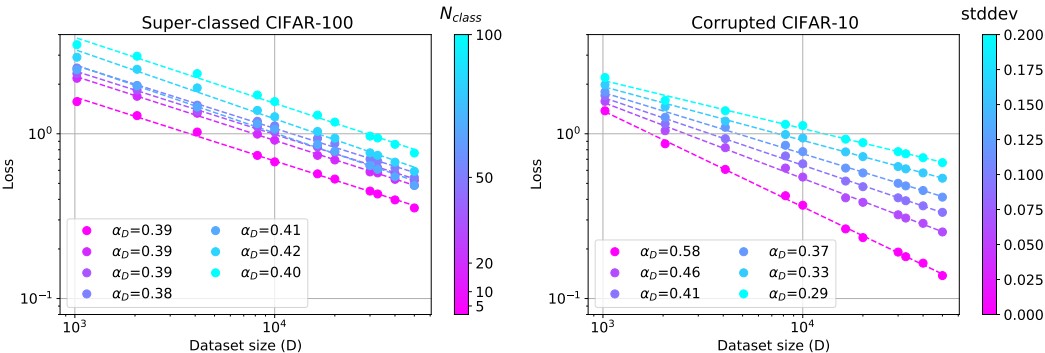

Figure 3: **Effect of data distribution on scaling exponents** For CIFAR-100 superclassed to $N$ classes **(left)**, we find that the number of target classes does not have a visible effect on the scaling exponent. **(right)** For CIFAR-10 with the addition of Gaussian noise to inputs, we find the strength of the noise has a strong effect on performance scaling with dataset size. All models are WRN-28-10.

**Data Manifolds and Kernels**  In Section 2.2, we discussed a simple argument that resolution-limited exponents $\alpha \propto 1/d$, where $d$ is the dimension of the data manifold. Our goal now is to explain how this connects with the linearized models and kernels discussed above: how does the spectrum of eigenvalues of a kernel relate to the dimension of the data manifold?

The key point is that sufficiently *smooth* kernels must have an eigenvalue spectrum with a bounded tail. Specifically, a $C^t$ kernel on a $d$-dimensional space must have eigenvalues $\lambda_n \lesssim \frac{1}{n^{1+t/d}}$ (Kühn, 1987). In the generic case where the covariance matrices we have discussed can be interpreted as kernels on a manifold, and they have spectra *saturating* the bound, linearized models will inherit scaling exponents given by the dimension of the manifold.

As a simple example, consider a $d$-torus. In this case we can study the Fourier series decomposition, and examine the case of a kernel $K(x - y)$. This must take the form $K = \sum_{n_I} [a_{n_I} \sin(n_I \cdot (x - y)) + b_{n_I} \cos(n_I \cdot (x - y))]$, where $n_I = (n_1, \cdots, n_d)$ are integer indices, and $a_{n_I}, b_{n_I}$ are the overall Fourier coefficients. To guarantee that $K$ is a $C^t$ function, we must have $a_{n_I}, b_{n_I} \lesssim \frac{1}{n^{d+t}}$ where $n^d = N$ indexes the number of $a_{n_I}$ in decreasing order. But this means that in this simple case, the tail eigenvalues of the kernel must be bounded by $\frac{1}{N^{1+t/d}}$ as $N \to \infty$.

## 2.4 DUALITY

We argued above that for kernels with pure power law spectra, the asymptotic scaling of the under-parameterized loss with respect to model size and the over-parameterized loss with respect to dataset size share a common exponent. In the linear setup at hand, the relation between the under-parameterized parameter dependence and over-parameterized dataset dependence is even stronger. The under-parameterized and over-parameterized losses are directly related by exchanging the projection onto random features with the projection onto random training points. Note, sample-wise double descent observed in Nakkiran (2019) is a concrete realization of this duality for a simple data distribution. In the supplement, we present examples exhibiting the duality of the loss dependence on model and dataset size outside of the asymptotic regime.

## 3 EXPERIMENTS

### 3.1 DEEP TEACHER-STUDENT MODELS

Our theory can be tested very directly in the teacher-student framework, in which a *teacher* deep neural network generates synthetic data used to train a *student* network. Here, it is possible to generate unlimited training samples and, crucially, controllably tune the dimension of the data manifold. We

accomplish the latter by scanning over the dimension of the inputs to the teacher. We have found that when scanning over both model size and dataset size, the interpolation exponents closely match the prediction of $4/d$. The dataset size scaling is shown in Figure 1, while model size scaling experiments appear in the supplement and have previously been observed in Sharma & Kaplan (2020).

### 3.2 VARIANCE-LIMITED SCALING IN THE WILD

Variance-limited scaling, (Section 2.1), can be universally observed in real datasets. Figure 1a (top-left, bottom-right) measures the variance-limited dataset scaling exponent $\alpha_D$ and width scaling exponent $\alpha_W$. In both cases, we find striking agreement with the theoretically predicted values $\alpha_D, \alpha_W = 1$ across a variety of dataset, network architecture, stochastic batch size and loss type. Our testbed includes deep fully-connected and convolutional networks with Relu or Erf nonlinearities and MSE or softmax-cross-entropy losses. The supplement contains experimental details.

### 3.3 RESOLUTION-LIMITED SCALING IN THE WILD

In addition to teacher-student models, we explored resolution-limited scaling behavior in the context of standard classification datasets. Wide ResNet (WRN) models (Zagoruyko & Komodakis, 2016) were trained for a fixed number of steps with cosine decay. In Figure 1b we also include data from a four hidden layer CNN detailed in the supplement. As detailed above, we find dataset dependent scaling behavior in this context.

We further investigated the effect of the data distribution on the resolution-limited exponent, $\alpha_D$, by tuning the number of target classes and input noise (Figure 3). To probe the effect of the number of classes, we constructed tasks derived from CIFAR-100 by grouping classes into broader semantic categories. We found that performance depends on the number of categories, but $\alpha_D$ is insensitive to this number. In contrast, the addition of Gaussian noise had a more pronounced effect on $\alpha_D$. This suggest a picture in which the network learns to model the input data manifold, independent of the classification task, consistent with observations in Nakkiran & Bansal (2020); Grathwohl et al. (2020).

We also explored the effect of aspect ratio on dataset scaling, finding that the exponent magnitude increases with width up to a critical width, while the dependence on depth is milder (see supplement).

## 4 DISCUSSION

We have presented a framework for categorizing neural scaling laws, along with derivations that help to explain their very general origins. Crucially, our predictions agree with empirical findings in settings which have often proven challenging for theory – deep neural networks on real datasets. The variance-scaling regime yields, for smooth test losses, a universal prediction of $\alpha_D = 1$ (for $D \gg P$) and $\alpha_W = 1$ (for $w \gg D$). The resolution-limited regime yields exponents whose numerical value is variable and data and model dependent.

There are many intriguing directions for future work; amongst these, we highlight one in particular. The invariance of the dataset scaling exponent to superclassing (Figure 3) suggests that deep networks may be largely learning properties of the input data manifold – akin to unsupervised learning – rather than significant task-specific structure, which may shed light on the versatility of learned deep network representations for different downstream tasks. This begs to be explored further.

**Limitations** One limitation is that our theoretical results are asymptotic, while experiments are performed with finite models and datasets. This is apparent in the resolution-limited regime which requires a hierarchy ($D \gg P$ or $P \gg D$). In Figures 1a and 2a top-right (bottom-left), we see a breakdown of the predicted scaling behavior as $D$ ($P$) become large and the hierarchy is lost. Furthermore in the resolution-limited regime for deep networks, our theoretical tools rely on positing the existence of a data manifold. A precise definition of the data manifold, however, is lacking forcing us to use imperfect proxies, such as nearest neighbor distances of final embedding layers.

**Ethics Statement** Work on scaling laws provides an opportunity for discussion on how to define and measure progress in machine learning. The values of exponents allow us to estimate expected gains that come from increases in scale of dataset, model, and compute. Applying similar considerations to other metrics (i.e. transfer, bias, robustness) in principle allows one to quantify whether and how models are improving or degrading with scale and at what environmental or computational cost. On the other hand, one may require that truly non-trivial progress in machine learning be progress that occurs *modulo scale*: namely, improvements in performance across different tasks that are not simple extrapolations of existing behavior. And perhaps the right combinations of algorithmic, model, and dataset improvements can lead to *emergent* behavior at new scales. Large language models such as GPT-3 (Fig. 1.2 in Brown et al. (2020)) have exhibited this in the context of few-shot learning. We hope our work spurs further research in understanding and controlling neural scaling laws.

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

## A   EXPERIMENTAL SETUP

**Figure 1 (top-left)**

Experiments utilize relatively small models, with the number of trainable parameteters $P \sim \mathcal{O}(1000)$, trained with full-batch gradient descent (GD) and small learning rate on datasets of size $D \gg P$. Each data point in the figure represents an average over subsets of size $D$ sampled from the full dataset. Experiments are done using Neural Tangents (Novak et al., 2020) based on JAX (Bradbury et al., 2018). All experiment except denoted as (CNN), use 3-layer, width-8 fully-connected networks. CNN architecture used is Myrtle-5 network (Shankar et al., 2020) with 8 channels. Relu activation function with critical initialization (Schoenholz et al., 2017; Lee et al., 2018; Xiao et al., 2018) was used. Unless specified softmax-cross-entropy loss was used. We performed full-batch gradient descent update for all dataset sizes without L2 regularization. 20 different training data sampling seeds were averaged for each point. For fully-connected networks, input pooling of size 4 was performed for CIFAR-10/100 dataset and pooling of size 2 was performed for MNIST and Fashion-MNIST dataset. This was to reduce number of parameters in the input layer (# of pixels $\times$ width) which can be quite large even for small width networks.

**Figure 1 (top-right)** All experiments were performed using a Flax (Heek et al., 2020) implementation of Wide ResNet 28-10 (Zagoruyko & Komodakis, 2016), and performed using the Caliban experiment manager (Ritchie et al., 2020). Models were trained for 78125 total steps with a cosine learning rate decay (Loshchilov & Hutter, 2016) and an augmentation policy consisting of random flips and crops. We report final loss, though we found no qualitative difference between using final loss, best loss, final accuracy or best accuracy (see Figure S1).

**Figure 1 (bottom-left)** The setup was identical to Figure 1 (top-right) except that the model considered was a depth 10 residual network with varying width.

**Figure 1 (bottom-right)**

Experiments are done using Neural Tangents. All experiments use 100 training samples and two-hidden layer fully-connected networks of varying width (ranging from $w = 64$ to $W = 11,585$) with Relu nonlinearities unless specified as Erf. Full-batch gradient descent and cross-entropy loss were used unless specified as MSE, and the figure shows curves from a random assortment of training times ranging from 100 to 500 steps (equivalently, epochs). Training was done with learning rates small enough so as to avoid catapult dynamics (Lewkowycz et al., 2020) and no $L2$ regularization; in such a setting, the infinite-width learning dynamics is known to be equivalent to that of linearized models (Lee et al., 2019). Consequently, for each random initialization of the parameters, the test loss of the finite-width linearized model was additionally computed in the identical training setting. This value approximates the limiting behavior $L(\infty)$ known theoretically and is subtracted off from the final test loss of the (nonlinear) neural network before averaging over 50 random initializations to yield each of the individual data points in the figure.

### A.1   DEEP TEACHER-STUDENT MODELS

The teacher-student scaling with dataset size (figure S2) was performed with fully-connected teacher and student networks with two hidden layers and widths 96 and 192, respectively, using PyTorch (Paszke et al., 2019). The inputs were random vectors sampled uniformly from a hypercube of dimension $d = 2, 3, \cdots, 9$. To mitigate noise, we ran the experiment on eight different random seeds, fixing the random seed for the teacher and student as we scanned over dataset sizes. We also used a fixed test dataset, and a fixed training set, which was sub-sampled for the experiments with smaller $D$. The student networks were trained using MSE loss and Adam optimizer with a maximum learning rate of $3 \times 10^{-3}$, a cosine learning rate decay, and a batch size of 64, and 40,000 steps of training. The test losses were measured with early stopping. We combine test losses from different random seeds by averaging the logarithm of the loss from each seed.

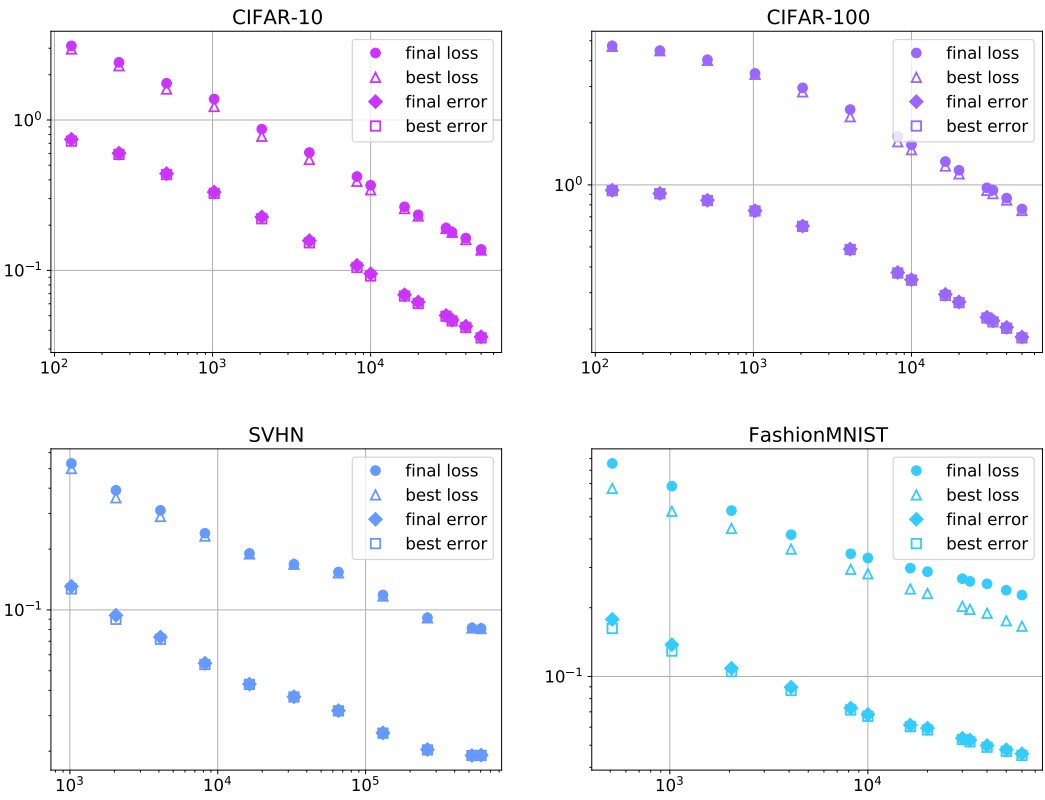

Figure S1: **Alternate metrics and stopping conditions** We find similar scaling behavior for both the loss and error, and for final and best (early stopped) metrics.

In our experiments, we always use inputs that are uniformly sampled from a $d$-dimensional hypercube, following the setup of Sharma & Kaplan (2020). They also utilized several intrisic dimension (ID) estimation methods and found the estimates were close to the input dimension, so we simply use the latter for comparisons. For the dataset size scans we used randomly initialized teachers with width 96, and students with width 192. We found similar results with other network sizes.

The final scaling exponents and input dimensions are show in the bottom of Figure 1b. We used the same experiments for the top of that figure, interpolating the behavior of both teacher and a set of students between two fixed training points. The students only differed by the size of their training sets, but had the same random seeds and were trained in the same way. In that figure the input space dimension was four.

Finally, we also used a similar setup to study variance-limited exponents and scaling. In that case we used much smaller models, with 16-dimensional hidden layers, and a correspondingly larger learning rate. We then studied scaling with $D$ again, with results pictured in Figure 1a.

## A.2  CNN ARCHITECTURE FOR RESOLUTION-LIMITED SCALING

Figure 1b includes data from CNN architectures trained on image datasets. The architectures are summarized in Table 1. We used Adam optimizer for training, with cross-entropy loss. Each network was trained for long enough to achieve either a clear minimum or a plateau in test loss. Specifically, CIFAR10, MNIST and Fashion MNIST were trained for 50 epochs, CIFAR100 was trained for 100 epochs and SVHN was trained for 10 epochs. The default Keras training parameters were used. In case of SVHN we included the additional images as training data. We averaged (in log space) over

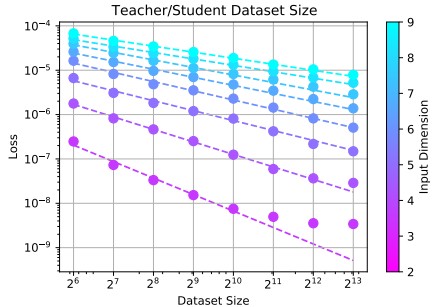

Figure S2: This figure shows scaling trends of MSE loss with dataset size for teacher/student models. The exponents extracted from these fits and their associated input-space dimensionalities are shown in Figure 1.

| Layer | Width |
|---|---|
| CNN window (3, 3) | 50 |
| 2D Max Pooling (2, 2) | |
| CNN window (3, 3) | 100 |
| 2D Max Pooling (2, 2) | |
| CNN window (3, 3) | 100 |
| Dense | 64 |
| Dense | 10 |

| Layer | Width |
|---|---|
| CNN window (3, 3) | 50 |
| 2D Max Pooling (2, 2) | |
| CNN window (3,3) | 100 |
| 2D Max Pooling (2, 2) | |
| CNN window (3, 3) | 200 |
| Dense | 256 |
| Dense | 100 |

| Layer | Width |
|---|---|
| CNN window (3, 3) | 64 |
| 2D Max Pooling (2, 2) | |
| CNN window (3, 3) | 64 |
| 2D Max Pooling (2, 2) | |
| Dense | 128 |
| Dense | 10 |

Table 1: CNN architectures for CIFAR10, MNIST, Fashion MNIST (left), CIFAR100 (center) and SVHN (right)

20 runs for CIFAR100 and CIFAR10, 16 runs for MNIST, 12 runs for Fashion MNIST, and 5 runs for SVHN. The results of these experiments are shown in Figure S3.

The measurement of input-space dimensionality for these experiments was done using the nearest-neighbour algorithm, described in detail in Appendix B and C in Sharma & Kaplan (2020). We used 2, 3 and 4 nearest neighbors and averaged over the three.

### A.3 TEACHER-STUDENT EXPERIMENT FOR SCALING OF LOSS WITH MODEL SIZE

We replicated the teacher-student setup in Sharma & Kaplan (2020) to demonstrate the scaling of loss with model size. The resulting variation of $-4/\alpha_P$ with input-space dimensionality is shown in figure S4. In our implementation we averaged (in log space) over 15 iterations, with a fixed, randomly generated teacher.

## B EFFECT OF ASPECT RATIO ON SCALING EXPONENTS

We trained Wide ResNet architectures of various widths and depths on CIFAR-10 accross dataset sizes. We found that the effect of depth on dataset scaling was mild for the range studied, while the effect of width impacted the scaling behavior up until a saturating width, after which the scaling behavior fixed. See Figure S5.

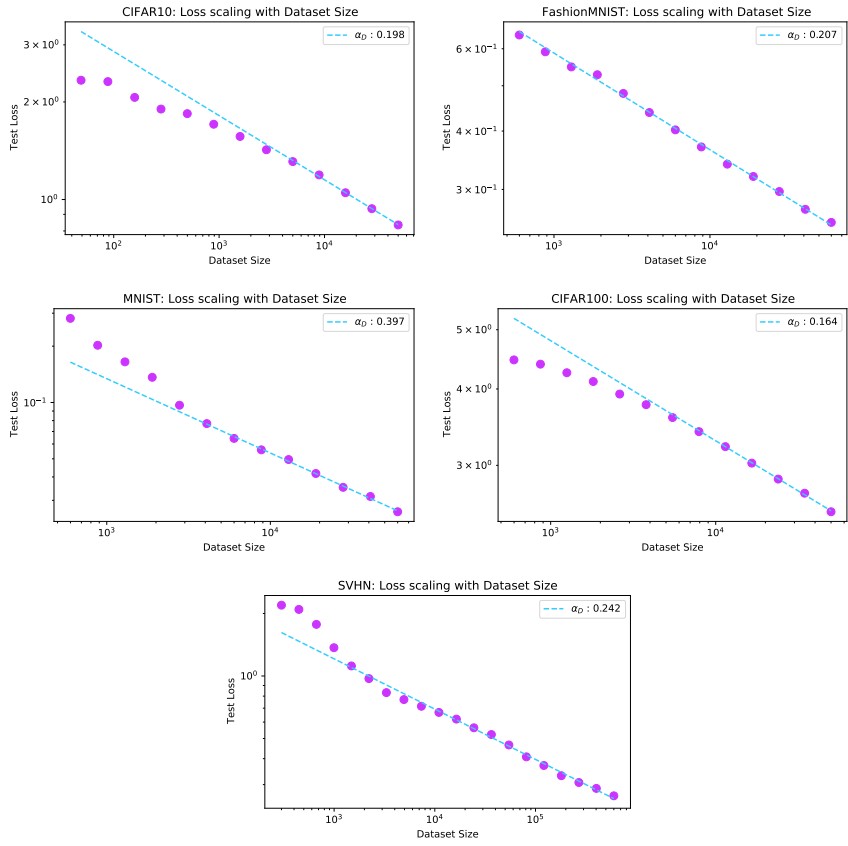

Figure S3: This figure shows scaling trends of CE loss with dataset size for various image datasets. The exponents extracted from these fits and their associated input-space dimensionalities are shown in Figure 1.

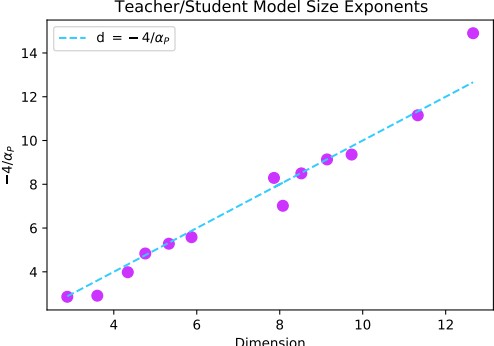

Figure S4: This figure shows the variation of $\alpha_P$ with the input-space dimension. The exponent $\alpha_P$ is the scaling exponent of loss with model size for teacher-student setup.

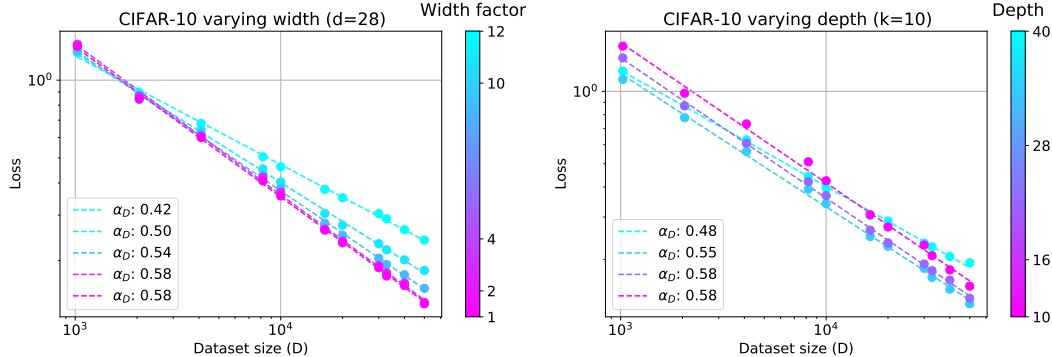

Figure S5: **Effect of aspect ratio on dataset scaling** We find that for WRN-d-k trained on CIFAR-10, varying depth from 10 to 40 has a relatively mild effect on scaling behavior, while varying the width multiplier, $k$, from 1 to 12 has a more noticeable effect, up until a saturating width.

## C    PROOF OF THEOREM 1

We now prove Theorem 1 repeated below for convenience.

**Theorem 1.** *Let $\ell(f)$ be the test loss as a function of network output, $(L = \mathbb{E}[\ell(f)])$, and let $f_T$ be the network output after $T$ training steps, thought of as a random variable over weight initialization, draws of the training dataset, and optimization seed. Further let $f_T$ be concentrating with $\mathbb{E}[(f_T - \mathbb{E}[f_T])^k] = \mathcal{O}(\epsilon) \forall k \geq 2$. If $\ell$ is a finite degree polynomial, or has bounded second derivative, or is 2-Hölder, then $\mathbb{E}[\ell(f_T)] - \ell(\mathbb{E}[f_T]) = \mathcal{O}(\epsilon)$.*

*Proof. Case 1 – finite degree polynomial*: In this case, we can write,

$$\ell(f_T) - \ell(\mathbb{E}[f_T]) = \sum_{k=1}^{K} \frac{\ell^{(k)}(\mathbb{E}[f_T])}{k!}(f_T - \mathbb{E}[f_T])^k \ , \tag{S1}$$

where $K$ is the polynomial degree and $\ell^{(k)}$ is the $k$-th derivative of $\ell$. Taking the expectation of (S1) and using the moment scaling proves the result.

*Case 2 – bounded second derivative*: The quadratic mean value theorem states that for any $f_T$, there exists a $c$ such that,

$$\ell(f_T) - \ell(\mathbb{E}[f_T]) = (f_T - \mathbb{E}[f_T])\ell'(\mathbb{E}[f_T]) + \frac{1}{2}\ell''(c)(f_T - \mathbb{E}[f_T])^2 \ . \tag{S2}$$

Taking the expectation of (S2) and using the fact that $f''(c)$ is bounded yields the desired result.

*Case 3 – 2-Hölder*: Lastly, the loss being 2-Hölder means we may write,

$$\ell(f_T) - \ell(\mathbb{E}[f_T]) \leq |\ell(f_T) - \ell(\mathbb{E}[f_T])| \leq K_\ell(f_T - \mathbb{E}[f_T])^2 \ . \tag{S3}$$

Again, taking the expectation of this inequality completes the proof. $\square$

**A note on loss variance**    Theorem 1 concerns the mean loss, however we would also like to understand if this scaling holds for typical instances. This can be understood by examining how the variance of the loss or alternatively how $\mathbb{E}[|\ell(f_T) - \ell(\mathbb{E}[f_T])|]$ scales.

For *Case 3 – 2-Hölder loss*, we can rerun the argument of Theorem 1, using (S3) to yield $\mathbb{E}[|\ell(f_T) - \ell(\mathbb{E}[f_T])|] = \mathcal{O}(\epsilon)$.

For *Cases 1 and 2*, we can attempt to apply the same argument as in the proof. This *almost* works. In particular, using Hölder's inequality, $\mathbb{E}[(f_T - \mathbb{E}[f_T])^k] = \mathcal{O}(\epsilon) \forall k \geq 2$ implies $\mathbb{E}[|f_T - \mathbb{E}[f_T]|^k] =$

$\mathcal{O}(\epsilon) \forall k \geq 2$. Taking the absolute value and expectation of (S1) or (S2) then gives

$$\mathbb{E}\left[|\ell(f_T) - \ell(\mathbb{E}[f_T])|\right] \leq |\ell'(\mathbb{E}[f_T])| \mathbb{E}\left[|f_T - \mathbb{E}[f_T]|\right] + \mathcal{O}(\epsilon). \tag{S4}$$

In general, the above assumptions on $\ell$ and $f_T$ imply only that $\mathbb{E}\left[|f_T - \mathbb{E}[f_T]|\right] = \mathcal{O}(\sqrt{\epsilon})$ and thus typical instances of the loss will exhibit a less dramatic scaling with $\epsilon$ than the mean. If we further assume, however, that $f_T$ on average has been trained such as to be sufficiently close to a local minimum of the loss, such that $|\ell'(\mathbb{E}[f_T])| = \mathcal{O}(\sqrt{\epsilon})$, then typical instances will also obey the $\mathcal{O}(\epsilon)$ scaling.

## D  VARIANCE-LIMITED DATASET SCALING

In this section, we expand on our discussion of the variance-limited dataset scaling, $L(D) - \lim_{D \to \infty} L(D) = \mathcal{O}(D^{-1})$. We first explain some intuition for why this behavior might be expected for sufficiently smooth loss. We then derive it explicitly for losses that are polynomial in the weights. Finally, we present non-smooth examples where the scaling can be violated either by having unbounded loss, or first derivative.

### D.1  INTUITION

At a high level, the intuition is as follows. For any fixed value of weights, $\theta$, the training loss with $D$ training points (thought of as a random variable over draws of the dataset), $L_{\text{train}}[\theta]$ concentrates around the population loss $L_{\text{pop}}[\theta]$, with variance that scales as $\mathcal{O}(D^{-1})$.

Our optimization procedure can be thought of as a map from initial weights and training loss to final weights $\text{Op}: (\theta_0, L_{\text{train}}[\theta]) \to \theta_T$. If this map is sufficiently smooth – for instance satisfying the assumptions of Theorem 1 or well approximated by a Taylor series about all $\mathbb{E}_D[L_{\text{train}}[\theta_t]]$ – then the output, $\theta_T$, will also concentrate around its infinite $D$ limit with variance scaling as $\mathcal{O}(D^{-1})$. Finally, if the population loss is also sufficiently smooth, the test loss for a model trained on $D$ data points averaged over draws of the dataset, $L(D) = \mathbb{E}_D[L_{\text{pop}}[\theta_T]]$, satisfies $L(D) - \lim_{D \to \infty} L(D) = \mathcal{O}(D^{-1})$. We now walk through this in a little more detail.

**Early time**  We can follow this intuition a bit more explicitly for the first few steps of gradient descent. As the training loss at initialization, $L_{\text{train}}[\theta_0]$, is a sample average over $D$ i.i.d draws, it concentrates around the population loss $L_{\text{pop}}[\theta_0]$ with variance $\mathcal{O}(D^{-1})$. As a result, the initial gradient, $g_0 = \frac{\partial L_{\text{train}}}{\partial \theta_0}$ will also concentrate with $O(D^{-1})$ variance and so will the weights at time 1, $\theta_1 = \theta_0 - \eta g_0$. The training loss at time step 1, is then given by

$$L_{\text{train}}[\theta_1] = L_{\text{train}}[\theta_0 - g_0]. \tag{S5}$$

If $L_{\text{train}}$ is sufficiently smooth around $\theta_0 - \mathbb{E}_D[g_0]$, then we get that $L_{\text{train}}[\theta_1]$ concentrates around $L_{\text{train}}[\theta_1]$ with $O(D^{-1})$ variance. We can keep bouncing back and forth between gradient (or equivalently weights) and training loss for any number of steps $T$ which does not scale with $D$. Plugging this final $\theta_T$ into the population loss and taking the expectation over draws of the training set, $L(D) = \mathbb{E}_D[L_{\text{pop}}[\theta_T]]$. If $L_{\text{pop}}$ is also sufficiently smooth, this yields $L(D) - \lim_{D \to \infty} L(D) = \mathcal{O}(D^{-1})$.

Here we have used the term sufficiently smooth. A sufficient set of criteria are given in Theorem 1; however this is likely too restrictive. Indeed, any set of train and population loss for which a Taylor series (or asymptomatic series with optimal truncation) give an $O(D^{-1})$ error around the training points $\mathbb{E}_D[\theta_{t=0...T}]$ will have this behavior.

**Local minimum**  The above intuition relied on training for a number of steps that was fixed as $D$ is taken large. Here we present some alternative intuition for the variance-limited scaling at late times, as training approaches a local minimum in the loss. For simplicity we discuss a one-dimensional loss.

Consider a local minimum, $\theta^*$, of the population loss. As $D$ is taken large, with high probability, the training loss will have a local minimum, $\bar{\theta}^*$, such that $|\theta^* - \bar{\theta}^*| = \mathcal{O}(D^{-1})$. One way to see this,

is to note that for a generic local minimum the first derivative changes sign, i.e. we can find $\theta_1, \theta_2$ such that $\theta_1 < \theta^* < \theta_2$ and either $L'_{\text{pop}}[\theta_1] < 0, L'_{\text{pop}}[\theta_2] > 0$ or $L'_{\text{pop}}[\theta_2] < 0, L'_{\text{pop}}[\theta_1] > 0$. To be concrete let's focus on the first case (the argument will be identical in either case). As $D$ becomes large, the probability that the training loss at $\theta_1$ and $\theta_2$ differs significantly from the population loss approaches zero. This can be seen from Markov's inequality, where, $P\left(\left|L'_{\text{train}}[\theta] - L'_{\text{pop}}[\theta]\right| > a\right) \leq \frac{\text{Var}_D\left(L'_{\text{train}}[\theta]\right)}{a^2}$, or more dramatically from Hoeffding's inequality (assuming bounded $L_{\text{train}} - L_{\text{pop}}$ lying in an interval of size $I$)

$$P\left(\left|L'_{\text{train}}[\theta] - L'_{\text{pop}}[\theta]\right| > a\right) \leq 2e^{-\frac{2}{I}D^2 a^2} . \tag{S6}$$

Here to have non-vanishing probability as we take $D$ large, $L'_{\text{train}}[\theta_1]$ and $L'_{\text{train}}[\theta_2]$ must be closer than $\mathcal{O}\left(D^{-1}\right)$. If $\theta_1$ and $\theta_2$ are taken to be $\mathcal{O}\left(D^0\right)$, then $L'_{\text{train}}$ must change sign, indicating an extremum of $L_{\text{train}}$; however we can do even better. If we assume $L_{\text{train}}$ is Lipshitz about $\theta^*$ then we can still ensure a sign change even if $|\theta_1 - \theta^*|, |\theta_2 - \theta^*| = \mathcal{O}\left(D^{-1}\right)$. Using concentration of $L''_{\text{train}}[\theta]$ ensures the extremum is a local minimum. For non-generic minimum (i.e. vanishing first derivatives) we can apply the same arguments to higher order derivatives (assuming they exist) of $L_{\text{pop}}$. Thus for a local minimum of $L_{\text{pop}}$, with high probability $L_{\text{train}}$ will have a corresponding minimum within a distance $\mathcal{O}\left(D^{-1}\right)$

If we now consider an initialization procedure, $\theta_0$, and training procedure such that training converges to the local minimum of the training loss, $\bar{\theta}^*$, and that the population loss is sufficiently smooth about $\theta^*$ (e.g. Lipshitz), then $\mathbb{E}_D[L_{\text{train}}[\bar{\theta}^*] - L_{\text{pop}}[\theta^*]] = \mathbb{E}_D[L_{\text{train}}[\bar{\theta}^*] - L_{\text{pop}}[\bar{\theta}^*]] + \mathbb{E}_D[L_{\text{pop}}[\bar{\theta}^*] - L_{\text{pop}}[\theta^*]]$. The first term vanishes, while the second is $\mathcal{O}(D^{-1})$. If we further assume that this happens on average over choices of $\theta_0$ then we expect $L(D) - \lim_{D\to\infty} L(D) = \mathcal{O}\left(D^{-1}\right)$.

**SGD** At first blush it may be surprising that the variance-limited scaling holds even for mini-batch training. Indeed in this case, there is batch noise that comes in at a much higher scale than any variance due to the finite training set size. Indeed, the effect of mini-batching changes the final test loss, however if we fix the SGD procedure or average over SGD seeds, as we take $D$ large, we can still ask how the training loss for a model trained under SGD on a training set of size $D$ differs from that for a model trained under SGD on an infinite training set.

To see this, we fist consider averaging over minibatches of size $B$, but where points are drawn i.i.d. with replacement. If we denote the batch at step $t$ by $\mathcal{B}_t$ and the average over independent draws of this batch by $\mathbb{E}_B[\bullet]$, then note we can translate moments with respect to batch draws with empirical averages over the entire training set. Explicitly, consider $c_a$ and $d_a$ potentially correlated, but each drawn i.i.d. within a batch. We have that,

$$\mathbb{E}_B\left[\frac{1}{B}\sum_{a\in\mathcal{B}_t} c_a\right] = \frac{1}{D}\sum_{a=1}^{D} c_a$$

$$\mathbb{E}_B\left[\left(\frac{1}{B}\sum_{a\in\mathcal{B}_t} c_a\right)\left(\frac{1}{B}\sum_{a'\in\mathcal{B}_t} d_{a'}\right)\right] = \left(1 - \frac{1}{B}\right)\left(\frac{1}{D}\sum_{a=1}^{D} c_a\right)\left(\frac{1}{D}\sum_{a'=1}^{D} d_{a'}\right) + \frac{1}{B}\frac{1}{D}\sum_{a=1}^{D} c_a d_a .$$
$$\tag{S7}$$

This procedure means, after taking an average over draws of SGD batch, rather than thinking about a function of mini-batch averages, we can equivalently consider a modified function, with explicit dependence on the batch size, but that is only a function of empirical means over the training set. We can thus recycle the above intuition for the scaling of smooth functions of empirical means.

The above relied on independently drawing every sample from every batch. At the other extreme, we can consider drawing batches without shuffling and increasing training set size by $B$ datapoints at a time, so as to keep the initial set of batches in an epoch fixed. In this case, the first deviation in training between a dataset of size $D$ and one of size $D + B$ happens at the last batch in the first epoch after processing $D$ datapoints.

As an extreme example, consider the case where $D > BT$. In this case, as we only take $T$ steps, the loss is constant for all $D > BT$ and so $\lim_{D \to \infty} L(D; T; B) = L(BT; T; B)$ and thus $L(D > BT) - \lim_{D \to \infty} L(D) = 0$ (and in particular is trivially $\mathcal{O}\left(D^{-1}\right)$).

## D.2 Polynomial loss

Before discussing neural network training we review the concentration behavior of polynomials of sample means.

**Lemma 1.** *Let* $\bar{c}^{(i)} = \frac{1}{D} \sum_{a=1}^{D} c_a^{(i)}$ *for* $i = 0 \ldots J$ *be empirical means, over* $D$ *i.i.d. draws of* $c_a^{(i)}$ *and let* $c^{(i)}$ *denote the distributional mean. Further, let* $X = (\bar{c}^{(0)})^{k_0} (\bar{c}^{(1)})^{k_1} \cdots (\bar{c}^{(J)})^{k_J}$ *be a monomial in the sample means. Then* $X$ *concentrates with moments* $\mathcal{O}\left(D^{-1}\right)$,

$$\mathbb{E}_D \left[ \left( X - (c^{(0)})^{k_0} (c^{(1)})^{k_1} \cdots (c^{(J)})^{k_J} \right)^n \right] = \mathcal{O}\left(D^{-1}\right). \tag{S8}$$

*Here,* $\mathbb{E}_D \left[ \bullet \right]$ *denotes the average over independent draws of* $D$ *samples.*

*Proof.* To establish this we can proceed by direct computation.

$$\mathbb{E}_D \left[ \left( X - (c^{(0)})^{k_0} (c^{(1)})^{k_1} \cdots (c^{(J)})^{k_J} \right)^n \right]$$
$$= \sum_{p=0}^{n} (-1)^{n-p} \binom{n}{p} \mathbb{E}_D \left[ X^p \right] \left( (c^{(0)})^{k_0} (c^{(1)})^{k_1} \cdots (c^{(J)})^{k_J} \right)^{n-p} \tag{S9}$$

Each term in the sum can be computed using

$$\mathbb{E}_D \left[ X^p \right] = \mathbb{E}_D \left[ (\bar{c}^{(0)})^{pk_0} (\bar{c}^{(1)})^{pk_1} \cdots (\bar{c}^{(J)})^{pk_J} \right]$$

$$= \frac{1}{D^{(p \sum_{i=0}^{J} k_i)}} \sum_{\{a_\alpha^{(i)}\}} \mathbb{E}_D \left[ \left( c_{a_1^{(0)}}^{(0)} \cdots c_{a_{pk_0}^{(0)}}^{(0)} \right) \left( c_{a_1^{(1)}}^{(1)} \cdots c_{a_{pk_1}^{(1)}}^{(1)} \right) \cdots \left( c_{a_1^{(J)}}^{(J)} \cdots c_{a_{pk_J}^{(J)}}^{(J)} \right) \right]$$

$$= \frac{1}{D^{(p \sum_{i=0}^{J} k_i)}} \sum_{\{a_\alpha^{(i)} \neq a_\beta^{(j)}\}} \mathbb{E}_D \left[ \left( c_{a_1^{(0)}}^{(0)} \cdots c_{a_{pk_0}^{(0)}}^{(0)} \right) \left( c_{a_1^{(1)}}^{(1)} \cdots c_{a_{pk_1}^{(1)}}^{(1)} \right) \cdots \left( c_{a_1^{(J)}}^{(J)} \cdots c_{a_{pk_J}^{(J)}}^{(J)} \right) \right]$$
$$+ \mathcal{O}\left(D^{-1}\right)$$

$$= \frac{D(D-1) \cdots (D - (p \sum_{i=0}^{J} k_i - 1))}{D^{(p \sum_{i=0}^{J} k_i)}} \left( c^{(0)} \right)^{pk_0} \left( c^{(1)} \right)^{pk_1} \cdots \left( c^{(J)} \right)^{pk_J} + \mathcal{O}\left(D^{-1}\right)$$

$$= \left( \left( c^{(0)} \right)^{k_0} \left( c^{(1)} \right)^{k_1} \cdots \left( c^{(J)} \right)^{k_J} \right)^p + \mathcal{O}\left(D^{-1}\right).$$

Plugging this into (S9) establishes the lemma. $\qquad \square$

In the above, we use the multi-index notation $\{a_\alpha^{(i)}\}$ for the collection of indices on the $c^i$ and the notation $\{a_\alpha^{(i)} \neq a_\beta^{(j)}\}$ for the subset of terms in the sum where all indices take different values.

Lemma 1 immediately implies that the mean of polynomials of $\bar{c}^{(i)}$ concentrate around their infinite data limit.

$$\mathbb{E}_D \left[ \left( g\left( \bar{c}^{(0)}, \bar{c}^{(1)}, \ldots, \bar{c}^{(K)} \right) - g\left( c^{(0)}, c^{(1)}, \ldots, c^{(K)} \right) \right)^n \right] = \mathcal{O}\left(D^{-1}\right), \tag{S10}$$

for $g \in P_K \left[ \bar{c}^{(0)}, \bar{c}^{(1)}, \ldots, \bar{c}^{(K)} \right]$.

With this out of the way, we can proceed to analysing the scaling of trained neural networks. Here we consider the simplified setting where the network map, $f$, and loss $\ell$ evaluated on each training example, $\mathbf{x}_a = (x_a, y_a)$, are polynomial of degree $J$ and $K$ in the weights, $\theta_\mu$,

$$f(x) = \sum_{i=1}^{J} b_{\mu_1 \mu_2 \ldots \mu_i}^{(i)}(x) \theta_{\mu_1} \theta_{\mu_2} \cdots \theta_{\mu_i} \quad \ell(\mathbf{x}_a) = \sum_{i=1}^{K} c_{\mu_1 \mu_2 \ldots \mu_i}^{(i)}(\mathbf{x}_a) \theta_{\mu_1} \theta_{\mu_2} \cdots \theta_{\mu_i}. \tag{S11}$$

The training loss can then be written as,

$$L_{\text{train}} = \sum_{i=1}^{K} \bar{c}_{\mu_1\mu_2\ldots\mu_i}^{(i)} \theta_{\mu_1}\theta_{\mu_2}\cdots\theta_{\mu_i}, \quad \bar{c}^{(i)} = \frac{1}{D}\sum_{a=1}^{D} c^{(i)}(\mathbf{x}_a). \tag{S12}$$

Here we have used the convention that the repeated weight indices $\mu_j$ are summed over.

**Gradient Descent**   As a result of the gradient descent weight update, $\theta_{t+1} = \theta_t - \eta\frac{\partial L_{\text{train}}}{\partial\theta}$, the weights at time $T$ are a polynomial of degree $(K-1)^T$ in the $\bar{c}^{(i)}$.

$$\theta_T \in P_{(K-1)^T}\left[\bar{c}^{(0)}, \bar{c}^{(1)}, \ldots, \bar{c}^{(K)}\right]. \tag{S13}$$

The coefficients of this polynomial depend on the initial weights, $\theta_0$. Plugging these weights back into the network output, we have that the network function at time $T$ is again a polynomial in $\bar{c}^{(i)}$, now with degree $J(K-1)^T$.

$$f_T(x) \in P_{J(K-1)^T}\left[\bar{c}^{(0)}, \bar{c}^{(1)}, \ldots, \bar{c}^{(K)}\right]. \tag{S14}$$

Thus, again using Lemma 1, $f_T$ concentrates with variance $\mathcal{O}(D^{-1})$.

$$\mathbb{E}_D\left[(f_T - \mathbb{E}_D[f_T])^2\right] = \mathcal{O}\left(D^{-1}\right). \tag{S15}$$

and by Theorem 1 the loss will obey they variance-limited scaling.

**Stochastic Gradient Descent**   We now consider the same setup of polynomial loss, but now trained via stochastic gradient descent (SGD). We consider SGD batches drawn i.i.d. with replacement and are interested in the test loss averaged over SGD draws, with fixed batch size, $B$.

We proceed by proving the following lemma, which allows us to reuse a similar argument to the GD case.

**Lemma 2.** *Let $\tilde{c}^{(i;t)} = \frac{1}{B}\sum_{a\in\mathcal{B}_t} c_a^{(i)}$ for $i = 0\ldots J$ be mini-batch averages, over $B$ i.i.d. draws of $c_a^{(i)}$. Further, let $X = (\tilde{c}^{(0;t_0)})^{k_0}(\tilde{c}^{(1;t_1)})^{k_1}\cdots(\tilde{c}^{(J;t_J)})^{k_J}$ be a monomial in the mini-batch means.*

*Then $\mathbb{E}_B[X] \in P_{\sum_{i=0}^{J} k_i}\left[\bar{d}^{(0)}, \bar{d}^{(1)}, \ldots, \bar{d}^{(\prod_{i=0}^{J}(k_i+1)-1)}\right]$, where $\bar{d}^{(i)}$ are empirical means over the full training set of i.i.d. random variables as in Lemma 1 and $\mathbb{E}_B[\bullet]$ denotes the expectation over draws of SGD batches of size $B$.*

*Proof.* Expectations over draws of batches at different time steps are independent. Thus, WLOG, we can consider $t := t_0 = t_1 = \cdots = t_J$. We can again proceed by direct computation, expanding the mini-batch sums.

$$\mathbb{E}_B[X] = \frac{1}{B^{\sum_{i=0}^{J} k_i}}\mathbb{E}_B\left[\sum_{\{a_\alpha^{(i)}\}\in\mathcal{B}_t}\left(c_{a_1^{(0)}}^{(0)}\cdots c_{a_{k_0}^{(0)}}^{(0)}\right)\left(c_{a_1^{(1)}}^{(1)}\cdots c_{a_{k_1}^{(1)}}^{(1)}\right)\cdots\left(c_{a_1^{(J)}}^{(J)}\cdots c_{a_{k_J}^{(J)}}^{(J)}\right)\right]. \tag{S16}$$

To proceed, we must keep track of terms in the sum where the $a_\alpha^{(i)}$ take the same or different values. If all $a_\alpha^{(i)}$ are different, the expectation over batch draws fully factorizes. More generally (S16) can be decomposed as a sum over products.

One way of keeping track of the index combinatorics is to introduce a set of graphs, $\Gamma$, where each graph $\gamma \in \Gamma$ has $k_0$ vertices of type 0, $k_1$ vertices of type 1, $\ldots$, and $k_J$ vertices of type J (one vertex for each $a_\alpha^{(i)}$ index). Any pair of vertices may have zero or one edge between them. For any set of three vertices, $v_1$, $v_2$, and $v_3$ with edges $(v_1, v_2)$ and $(v_2, v_3)$ there must also be an edge $(v_1, v_3)$. The set $\Gamma$ consists of all possible ways of connecting these vertices consistent with these rules.

For each graph, $\gamma$, we denote connected components by $\sigma$ and denote the number of vertices of type $i$ within the connected component $\sigma$ by $m_\sigma^{(i)}$. With this we can write the sum, (S16) as

$$
\begin{aligned}
\mathbb{E}_B[X] &= \sum_{\gamma \in \Gamma} S_\gamma(B) \prod_{\sigma \in \gamma} \mathbb{E}_B \left[ \frac{1}{B} \sum_{a \in \mathcal{B}_t} \left( c_a^{(0)} \right)^{m_\sigma^{(0)}} \left( c_a^{(1)} \right)^{m_\sigma^{(1)}} \cdots \left( c_a^{(J)} \right)^{m_\sigma^{(J)}} \right] \\
&= \sum_{\gamma \in \Gamma} S_\gamma(B) \prod_{\sigma \in \gamma} \frac{1}{D} \sum_{a=1}^{D} \left( c_a^{(0)} \right)^{m_\sigma^{(0)}} \left( c_a^{(1)} \right)^{m_\sigma^{(1)}} \cdots \left( c_a^{(J)} \right)^{m_\sigma^{(J)}} \\
&= \sum_{\gamma \in \Gamma} S_\gamma(B) \prod_{\sigma \in \gamma} \bar{d}^{(\{m_\sigma^{(0)}, m_\sigma^{(1)}, \dots, m_\sigma^{(J)}\})} \, .
\end{aligned}
\tag{S17}
$$

Here $S_\gamma(B)$ is a combinatoric factor associated to each graph, not relevant for the argument. The $m_\sigma^{(i)}$ take on values 0 to $k_i$, so the multi-index, can take on $\prod_{i=1}^{J}(k_i + 1)$ different values, which we re-index to $\bar{d}^{(0)}, \bar{d}^{(1)}, \dots, \bar{d}^{(\prod_{i=1}^{J}(k_i+1)-1)}$. Meanwhile, the degree of (S17) in $\bar{d}^{(i)}$ is bounded by the number of total vertices in each graph, i.e. $\sum_{i=0}^{J} k_i$. This establishes the lemma. $\qquad\square$

For a polynomial loss of degree $K$, the mini-batch training loss at each time step takes the form

$$
L_{\text{train}}^{(t)} = \sum_{i=1}^{K} \tilde{c}_{\mu_1 \mu_2 \dots \mu_i}^{(i;t)} \theta_{\mu_1} \theta_{\mu_2} \cdots \theta_{\mu_i} \, , \quad \tilde{c}^{(i;t)} = \frac{1}{B} \sum_{a=\in \mathcal{B}_t} c^{(i)}(\mathbf{x}_a) \, .
\tag{S18}
$$

The update rule, $\theta_{t+1} = \theta_t - \eta \frac{\partial L_{\text{train}}^{(t+1)}}{\partial \theta}$ ensures that $\theta_T$ is a polynomial of degree $(K-1)^T$ in the $\tilde{c}^{(i;0)}, \tilde{c}^{(i;1)}, \cdots, \tilde{c}^{(i;T)}$

$$
\theta_T \in P_{(K-1)^T} \left[ \tilde{c}^{(0;0)}, \tilde{c}^{(0;1)}, \dots, \tilde{c}^{(0;T)}, \tilde{c}^{(1;0)}, \tilde{c}^{(1;1)}, \dots, \tilde{c}^{(1;T)}, \dots, \tilde{c}^{(K;0)}, \tilde{c}^{(K;1)}, \dots, \tilde{c}^{(K;T)} \right] \, ,
\tag{S19}
$$

and consequently, denoting the test loss evaluated at $\theta_T$ by $L[\theta_T]$,

$$
L[\theta_T] \in P_{K(K-1)^T} \left[ \tilde{c}^{(0;0)}, \tilde{c}^{(0;1)}, \dots, \tilde{c}^{(0;T)}, \tilde{c}^{(1;0)}, \tilde{c}^{(1;1)}, \dots, \tilde{c}^{(1;T)}, \dots, \tilde{c}^{(K;0)}, \tilde{c}^{(K;1)}, \dots, \tilde{c}^{(K;T)} \right] \, .
\tag{S20}
$$

Using Lemma 2, the expectation of $L[\theta_T]$ over draws of SGD batches is given by

$$
\mathbb{E}_B[L[\theta_T]] \in P_{K(K-1)^T} \left[ \bar{d}^{(0)}, \dots, \bar{d}^{(K^K(K-1)^{TK})} \right] \, .
\tag{S21}
$$

Finally, denoting $\mathbb{E}_D[\mathbb{E}_B[L[\theta_T]]]$ by $L(D; B)$ and applying Lemma 1 gives

$$
L(D; B) - \lim_{D \to \infty} L(D; B) = \mathcal{O}\left(D^{-1}\right) \, .
\tag{S22}
$$

## D.3 Non-smooth examples

Here we present two worked examples where non-bounded or non-smooth loss leads to violations of the variance dominated scaling. In example one, the system obeys the variance dominated scaling at early times, but exhibits different behavior for times larger than the dataset size. In the second example, the system violates the variance dominated scaling even for two gradient descent steps, as a result of an unbounded derivative in the loss.[3]

**Example 1 – unbounded loss at late times**  Consider a dataset with two varieties of data points, drawn with probabilities $\alpha$ and $1 - \alpha$, and one-dimensional quadratic losses, $\ell_1$ (concave up) and $\ell_2$ (concave down), on these two varieties.

$$
\ell_1(\theta) = \frac{1}{2}\theta^2 \, , \quad \ell_2(\theta) = -\frac{1}{2}\theta^2 \, .
\tag{S23}
$$

---

[3] We thank Anonymous for suggesting these two types of examples.

If, in a slight abuse of notation, we further denote the training loss on a sample with $n_1$ points of type 1 and $D - n_1$ points of type two by $\ell_{n_1}$ and the population loss at a given value of the weight by $L_{\text{pop}}$, we have

$$\ell_{n_1} = \left(\frac{n_1}{D} - \frac{1}{2}\right)\theta^2\,, \quad L_{\text{pop}} = \left(\alpha - \frac{1}{2}\right)\theta^2\,. \tag{S24}$$

For this example we take $\alpha > 1/2$. In this case, the minimum of the population loss is at zero, while the minimum of the training loss can be at zero, or at $\pm\infty$ depending on whether the training sample has $n_1$ greater than or less than $D/2$. We can thus create a situation where at late training times, $\theta_T$ does not concentrate around the minimum of the population loss.

As we work through this example explicitly, we will see the following. (i) A mismatch larger than $\mathcal{O}\left(D^{-1}\right)$ between the population minimum and the minimum found by training on a sample set of size $D$ requires times $T$ larger than a constant multiple of $D$. (ii) The quantity we study throughout this work is the difference between the infinite data limit of the test loss, and the finite data value, $L(D) - \lim_{D\to\infty} L(D)$. The minimum of the infinite data limit of the test loss is not the same as the minimum of the population loss, $\min \lim_{D\to\infty} L(D) \neq \min_\theta L_{\text{pop}}$. In this example one diverges, while the other is finite. In particular this example evades the scaling result by $L(D)$ for times larger than $D$ having a diverging limit.

Explicitly, we study the evolution of the model under gradient flow.

$$\dot{\theta} = -2\left(\frac{n_1}{D} - \frac{1}{2}\right)\theta\,, \quad \theta_T = e^{-2\left(\frac{n_1}{D} - \frac{1}{2}\right)T}\theta_0\,. \tag{S25}$$

The test loss averaged over draws of the dataset is given by

$$L(D;T) = \mathbb{E}_{n_1}\left[\left(\alpha - \frac{1}{2}\right)\theta_T^2\right] = e^{2T}\left(\alpha - \frac{1}{2}\right)\left(1 - \alpha\left(1 - e^{-\frac{4T}{D}}\right)\right)^D \theta_0^2 \tag{S26}$$

If we consider this loss at large $D$ and fixed $T$ we get

$$L(D;T) = e^{-4\left(\alpha - \frac{1}{2}\right)T}\left(\alpha - \frac{1}{2}\right)\theta_0^2\left(1 + \frac{8T^2\alpha(1 - \alpha)}{D} + \mathcal{O}\left(D^{-2}\right)\right)\,, \tag{S27}$$

and thus $L(D;T) - \lim_{D\to\infty} L(D;T) = \mathcal{O}\left(D^{-1}\right)$ as expected.

If on the other hand we consider taking $T \gg D$ we have

$$L(D;T \gg D) = e^{2T}\left(\alpha - \frac{1}{2}\right)(1 - \alpha)^D \theta_0^2\,, \tag{S28}$$

the limit $\lim_{D,T\to\infty} L(D;T \gg D)$ diverges.

Lastly, we note that if we take $T = \beta D$ with $\beta < |\log(1 - \alpha)|/2$ we can approach the large $D$ limit with non-generic, tuneable exponential convergence.

**Example 2 – unbounded derivative**   Again, consider a two variety setup, this case with equal probabilities and per sample losses,

$$\ell_1(\theta) = \frac{1}{2}\theta^2 + \frac{1}{2\alpha}|\theta|^\alpha\,, \quad \ell_2(\theta) = \frac{1}{2}\theta^2 - \frac{1}{2\alpha}|\theta|^\alpha\,. \tag{S29}$$

We will consider different values of $\alpha > 0$. The train loss and population loss are then,

$$\ell_{n_1} = \frac{1}{2}\theta^2 + \frac{1}{\alpha}\left(\frac{n_1}{D} - \frac{1}{2}\right)|\theta|^\alpha\,, \quad L_{\text{pop}} = \frac{1}{2}\theta^2\,. \tag{S30}$$

We consider a model initialized to $\theta_0 = 1$ and trained for two steps of gradient descent with learning rate 1.

$$g_t = \theta_t + \left(\frac{n_1}{D} - \frac{1}{2}\right)\theta_t|\theta_t|^{\alpha - 2}\,, \quad \theta_{t+1} = \theta_t - g_t\,. \tag{S31}$$

Two update steps gives

$$\theta_2 = \left| \frac{n_1}{D} - \frac{1}{2} \right|^\alpha . \tag{S32}$$

The test loss is given by the population loss evaluated at $\theta_2$ averaged over test set draws.

$$
\begin{aligned}
L(D) = \mathbb{E}_{n_1} \left[ \frac{1}{2} \theta_2^2 \right] &= \frac{1}{2^{D+1}} \sum_{n_1=0}^{D} \binom{D}{n_1} \left| \frac{n_1}{D} - \frac{1}{2} \right|^{2\alpha} \\
&= \sqrt{\frac{D}{2\pi}} \int_{-\infty}^{\infty} e^{-2D\left(x-\frac{1}{2}\right)^2} \left| x - \frac{1}{2} \right|^{2\alpha} + \mathcal{O}\left(D^{-1}\right) \\
&= \frac{\Gamma\left(\alpha + \frac{1}{2}\right)}{2^{1+\alpha}\sqrt{\pi}} D^{-\alpha} + \mathcal{O}\left(D^{-1}\right) .
\end{aligned}
\tag{S33}
$$

Here we have approximated the binomial distribution at large $D$ with a normal distribution using Stirling's approximation.

Note that if $\alpha \geq 1$ then $L(D) - \lim_{D\to\infty} L(D) = \mathcal{O}\left(D^{-1}\right)$ i.e. the finite sample loss approaches the infinite data loss with the predicted variance-limited scaling. For $0 < \alpha < 1$, we get a different scaling controlled by $\alpha$. Note that the gradient, expression (S31) is singular at the origin for $\alpha$ precisely in this range.

In summary, this example achieves a different scaling exponent through a diverging gradient.

## E    PROOF OF THEOREMS 2 AND 3

In this section we detail the proof of Theorems 2 and 3. The key observation is to make use of the fact that nearest neighbor distances for $D$ points sampled i.i.d. from a $d$-dimensional manifold have mean $\mathbb{E}_{D,x}\left[\|x - \hat{x}\|\right] = \mathcal{O}\left(D^{-1/d}\right)$, where $\hat{x}$ is the nearest neighbor of $x$ and the expectation is the mean over data-points and draws of the dataset see e.g. (Levina & Bickel, 2005).

The theorem statements are copied for convenience. In the main, in an abuse of notation, we used $L(f)$ to indicate the value of the test loss as a function of the network $f$, and $L(D)$ to indicate the test loss averaged over the population, draws of the dataset, model initializations and training. To be more explicit below, we will use the notation $\ell(f(x))$ to indicate the test loss for a single network evaluated at single test point.

**Theorem 2.** *Let $\ell(f)$, $f$ and $\mathcal{F}$ be Lipschitz with constants $K_L$, $K_f$, and $K_{\mathcal{F}}$ and $\ell(\mathcal{F}) = 0$. Further let $\mathcal{D}$ be a training dataset of size $D$ sampled i.i.d from $\mathcal{M}_d$ and let $f(x) = \mathcal{F}(x)$, $\forall x \in \mathcal{D}$ then $L(D) = \mathcal{O}\left(K_L max(K_f, K_{\mathcal{F}}) D^{-1/d}\right)$.*

*Proof.* Consider a network trained on a particular draw of the training data. For each training point, $x$, let $\hat{x}$ denote the neighboring training data point. Then by the above Lipschitz assumptions and the vanishing of the loss on the true target, we have $\ell(f(x)) \leq K_L |f(x) - \mathcal{F}(x)| \leq K_L (K_f + K_{\mathcal{F}}) |x - \hat{x}|$. With this, the average test loss is bounded as

$$L(D) \leq K_L (K_f + K_{\mathcal{F}}) \mathbb{E}_{D,x}\left[\|x - \hat{x}\|\right] = \mathcal{O}\left(K_L max(K_f, K_{\mathcal{F}}) D^{-1/d}\right) . \tag{S34}$$

In the last equality, we used the above mentioned scaling of nearest neighbor distances.    $\square$

**Theorem 3.** *Let $\ell(f)$, $f$ and $\mathcal{F}$ be Lipschitz with constants $K_L$, $K_f$, and $K_{\mathcal{F}}$. Further let $f(x) = \mathcal{F}(x)$ for $P$ points sampled i.i.d from $\mathcal{M}_d$ then $L(P) = \mathcal{O}\left(K_L max(K_f, K_{\mathcal{F}}) P^{-1/d}\right)$.*

*Proof.* Denote by $\mathcal{P}$ the $P$ points, $z$, for which $f(z) = \mathcal{F}(z)$. For each test point $x$ let $\hat{x}$ denote the closest point in $\mathcal{P}$, $\hat{x} = \text{argmin}_{\mathcal{P}} (|x - z|)$. Adopting this notation, the result follows by the same argument as Theorem 2.    $\square$

# F   RANDOM FEATURE MODELS

Here we present random feature models in more detail. We begin by reviewing exact expressions for the loss. We then go onto derive its asymptotic properties. We again consider training a model $f(x) = \sum_{\mu=1}^{P} \theta_\mu f_\mu(x)$, where $f_\mu$ are drawn from some larger pool of features, $\{F_M\}$, $f_\mu(x) = \sum_{M=1}^{S} \mathcal{P}_{\mu M} F_M(x)$.

Note, if $\{F_M(x)\}$ form a complete set of functions over the data distribution, than any target function, $y(x)$, can be expressed as $y = \sum_{M=1}^{S} \omega_M F_M(x)$. The extra constraint in a teacher-student model is specifying the distribution of the $\omega_M$. The variance-limited scaling goes through with or without the teacher-student assumption, however it is crucial for analysing the variance-limited behavior.

As in Section 2.3 we consider models with weights initialized to zero and trained to convergence with mean squared error loss.

$$L_{\text{train}} = \frac{1}{2D} \sum_{a=1}^{D} (f(x_a) - y_a)^2 \,. \tag{S35}$$

The data and feature second moments play a central role in our analysis. We introduce the notation,

$$\mathcal{C} = \mathbb{E}_x \left[ F(x) F^T(x) \right], \quad \bar{\mathcal{C}} = \frac{1}{D} \sum_{a=1}^{D} F(x_a) F^T(x_a), \quad C = \mathcal{P} \mathcal{C} \mathcal{P}^T, \quad \bar{C} = \mathcal{P} \bar{\mathcal{C}} \mathcal{P}^T \,.$$
$$\mathcal{K}(x, x') = \frac{1}{S} F^T(x) F(x'), \quad \bar{\mathcal{K}} = \mathcal{K} \Big|_{\mathcal{D}_{\text{train}}}, \quad K(x, x') = \frac{1}{P} f^T(x) f(x'), \quad \bar{K} = K \Big|_{\mathcal{D}_{\text{train}}} \,. \tag{S36}$$

Here the script notation indicates the full feature space while the block letters are restricted to the student features. The bar represents restriction to the training dataset. We will also indicate kernels with one index in the training set as $\vec{\mathcal{K}}(x) := \mathcal{K}(x, x_{a=1\ldots D})$ and $\vec{K}(x) := K(x, x_{a=1\ldots D})$. After this notation spree, the test loss can be written for under-parameterized models, $P \leq D$ as

$$L(D, P) = \frac{1}{2S} \mathbb{E}_D \left[ \text{Tr} \left( \mathcal{C} + \bar{\mathcal{C}} \mathcal{P}^T \bar{C}^{-1} C \bar{C}^{-1} \mathcal{P} \bar{\mathcal{C}} - 2 \bar{\mathcal{C}} \mathcal{P}^T \bar{C}^{-1} \mathcal{P} \mathcal{C} \right) \right] \,. \tag{S37}$$

and for over-parameterized models (at the unique minimum found by GD, SGD, or projected Newton's method),

$$L(D, P) = \frac{1}{2} \mathbb{E}_{x, D} \left[ \mathcal{K}(x, x) + \vec{K}(x)^T \bar{K}^{-1} \bar{\mathcal{K}} \bar{K}^{-1} \vec{K}(x) - 2 \vec{K}(x)^T \bar{K}^{-1} \vec{\mathcal{K}}(x) \right] \,. \tag{S38}$$

Here the expectation $\mathbb{E}_D [\bullet]$ is an expectation with respect to iid draws of a dataset of size $D$ from the input distribution, while $\mathbb{E}_x [\bullet]$ is an ordinary expectation over the input distribution. Note, expression (S37) is also valid for over-parameterized models and (S38) is valid for under-parameterized models if the inverses are replaces with the Moore-Penrose pseudo-inverse. Also note, the two expressions can be related by echanging the projections onto finite features with the projection onto the training dataset and the sums of teacher features with the expectation over the data manifold. This realizes the duality between dataset and features discussed above.

## F.1   ASYMPTOTIC EXPRESSIONS

We are interested in (S37) and (S38) in the limits of large $P$ and $D$.

**Variance-limited scaling**   We begin with the under-parameterized case. In the limit of lots of data the sample estimate of the feature feature second moment matrix, $\bar{\mathcal{C}}$, approaches the true second moment matrix, $\mathcal{C}$. Explicitly, if we define the difference, $\delta \mathcal{C}$ by $\bar{\mathcal{C}} = \mathcal{C} + \delta \mathcal{C}$. We have

$$\mathbb{E}_D [\delta \mathcal{C}] = 0$$
$$\mathbb{E}_D [\delta \mathcal{C}_{M_1 N_1} \delta \mathcal{C}_{M_2 N_2}] = \frac{1}{D} \left( \mathbb{E}_x [F_{M_1}(x) F_{N_1}(x) F_{M_2}(x) F_{N_2}(x)] - \mathcal{C}_{M_1 N_1} \mathcal{C}_{M_2 N_2} \right) \tag{S39}$$
$$\mathbb{E}_D [\delta \mathcal{C}_{M_1 N_1} \cdots \delta \mathcal{C}_{M_n N_n}] = \mathcal{O} \left( D^{-2} \right) \quad \forall n > 2 \,.$$

The key takeaway from (S39) is that the dependence on $D$ is manifest.

Using these expressions in (S37) yields.

$$
\begin{aligned}
L(D, P) = &\frac{1}{2S} \operatorname{Tr} \left( \mathcal{C} - \mathcal{C} \mathcal{P}^T C^{-1} \mathcal{P} \mathcal{C} \right) \\
&+ \frac{1}{2DS} \sum_{M_{1,2} N_{1,2}=1}^{P} T_{M_1 N_1 M_2 N_2} \left[ \delta_{M_1 M_2} \left( \mathcal{P}^T C^{-1} \mathcal{P} \right)_{N_1 N_2} + (C^{-1} \mathcal{P} \mathcal{C}^2 \mathcal{P}^T C^{-1})_{M_1 M_2} C^{-1}_{N_1 N_2} \right. \\
&\left. \qquad\qquad - 2 \left( \mathcal{C} \mathcal{P}^T C^{-1} \mathcal{P} \right)_{M_1 M_2} \left( \mathcal{P}^T C^{-1} \mathcal{P} \right)_{N_1 N_2} \right] + \mathcal{O} \left( D^{-2} \right) .
\end{aligned}
\tag{S40}
$$

Here we have introduced the notation, $T_{M_1 N_1 M_2 N_2} = \mathbb{E}_x \left[ F_{M_1}(x) F_{N_1}(x) F_{M_2}(x) F_{N_2}(x) \right]$.

As above, defining

$$
L(P) := \lim_{D \to \infty} L(D, P) = \frac{1}{2S} \operatorname{Tr} \left( \mathcal{C} - \mathcal{C} \mathcal{P}^T C^{-1} \mathcal{P} \mathcal{C} \right) .
\tag{S41}
$$

we see that though $L(D, P) - L(P)$ is a somewhat cumbersome quantity to compute, involving the average of a quartic tensor over the data distribution, its dependence on $D$ is simple.

For the over-parameterized case, we can similarly expand (S38) using $K = \mathcal{K} + \delta\mathcal{K}$. With fluctuations satisfying,

$$
\mathbb{E}_P \left[ \delta\mathcal{K} \right] = 0
$$

$$
\mathbb{E}_P \left[ \delta\mathcal{K}_{a_1 b_1} \delta\mathcal{K}_{a_2 b_2} \right] = \frac{1}{P} \left( \mathbb{E}_P \left[ f_\mu(x_{a_1}) f_\mu(x_{b_1}) f_\mu(x_{a_2}) f_\mu(x_{b_2}) \right] - \mathcal{K}_{a_1 b_1} \mathcal{K}_{a_2 b_2} \right)
\tag{S42}
$$

$$
\mathbb{E}_P \left[ \delta\mathcal{K}_{a_1 a_1} \cdots \delta\mathcal{K}_{a_n a_n} \right] = \mathcal{O} \left( P^{-2} \right) \quad \forall n > 2 .
$$

This gives the expansion

$$
L(D, P) = \frac{1}{2} \mathbb{E}_{x,D} \left[ \mathcal{K}(x, x) - \vec{\mathcal{K}}(x)^T \bar{\mathcal{K}}^{-1} \vec{\mathcal{K}}(x) \right] + \mathcal{O}(P^{-1}) ,
\tag{S43}
$$

and

$$
L(D) = \frac{1}{2} \mathbb{E}_{x,D} \left[ \mathcal{K}(x, x) - \vec{\mathcal{K}}(x)^T \bar{\mathcal{K}}^{-1} \vec{\mathcal{K}}(x) \right] .
\tag{S44}
$$

**Resolution-limited scaling**   We now move onto studying the parameter scaling of $L(P)$ and dataset scaling of $L(D)$. We explicitly analyse the dataset scaling of $L(D)$, with the parameter scaling following via the dataset parameter duality.

Much work has been devoted to evaluating the expression, (S44) (Williams & Vivarelli, 2000; Malzahn & Opper, 2002; Sollich & Halees, 2002). One approach is to use the *replica trick* – a tool originating in the study of disordered systems which computes the expectation of a logarithm of a random variable via simpler moment contributions and analyticity assumption (Parisi, 1980). The replica trick has a long history as a technique to study the generalization properties of kernel methods (Sollich, 1998; Malzahn & Opper, 2001; 2003; Urry & Sollich, 2012; Cohen et al., 2019; Gerace et al., 2020; Bordelon et al., 2020). We will most closely follow the work of Canatar et al. (2021) who use the replica method to derive an expression for the test loss of linear feature models in terms of the eigenvalues of the kernel $\mathcal{C}$ and $\bar\omega$, the coefficient vector of the target labels in terms of the model features.

$$
L(D) = \frac{\kappa^2}{1 - \gamma} \sum_i \frac{\lambda_i \bar\omega_i^2}{\left( \kappa + D\lambda_i \right)^2} ,
$$

$$
\kappa = \sum_i \frac{\kappa \lambda_i}{\kappa + D\lambda_i} , \quad \gamma = \sum_i \frac{D\lambda_i^2}{\left( \kappa + D\lambda_i \right)^2} .
\tag{S45}
$$

This is the ridge-less, noise-free limit of equation (4) of Canatar et al. (2021). Here we analyze the asymptotic behavior of these expressions for eigenvalues satisfying a power-law decay, $\lambda_i = i^{-(1+\alpha_K)}$ and for targets coming from a teacher-student setup, $w \sim \mathcal{N}(0, 1/S)$.

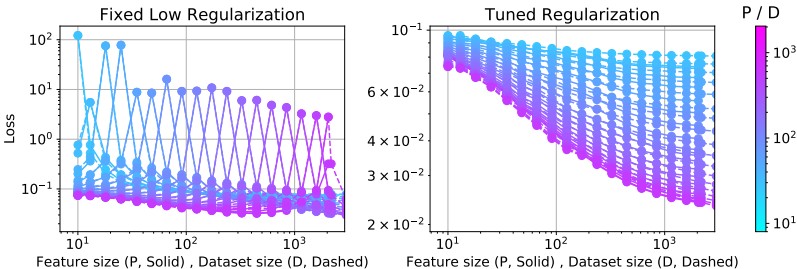

Figure S6: **Duality between dataset size vs feature number in pretrained features** Using pretrained embedding features of EfficientNet-B5 (Tan & Le, 2019) for different levels of regularization, we see that loss as function of dataset size or loss as a function of the feature dimension track each other both for small regularization (**left**) and for tuned regularization (**right**). Note that regularization strength with trained-feature kernels can be mapped to inverse training time (Ali et al., 2019; Lee et al., 2020). Thus (**left**) corresponds to long training time and exhibits double descent behavior, while (**right**) corresponds to optimal early stopping.

To begin, we note that for teacher-student models in the limit of many features, the overlap coefficients $\bar{\omega}$ are equal to the teacher weights, up to a rotation $\bar{\omega}_i = O_{iM} w_M$. As we are choosing an isotropic Gaussian initialization, we are insensitive to this rotation and, in particular, $\mathbb{E}_w \left[ \bar{\omega}_i^2 \right] = 1/S$. See Figure S8 for empirical support of the average constancy of $\bar{\omega}_i$ for the teacher-student setting and contrast with realistic labels.

With this simplification, we now compute the asymptotic scaling of (S45) by approximating the sums with integrals and expanding the resulting expressions in large $D$. We use the identities:

$$\int_1^\infty dx \frac{x^{-n(1+\alpha)}}{\left(\kappa + Dx^{-(1+\alpha)}\right)^m} = \kappa^{-m} \frac{\Gamma\left(n - \frac{1}{1+\alpha}\right)}{(1+\alpha)\Gamma\left(n + \frac{\alpha}{1+\alpha}\right)} {}_2F_1\left(m, n - \frac{1}{1+\alpha}, n + \frac{\alpha}{1+\alpha}, \frac{-D}{\kappa}\right)$$

$$ {}_2F_1\left(a, b, c, -y\right) \propto y^{-a} + \mathcal{B}y^{-b} + \dots ,$$

$$(S46)$$

Here ${}_2F_1$ is the hypergeometric function and the second line gives its asymptotic form at large y. $\mathcal{B}$ is a constant which does not effect the asymptotic scaling.

Using these relations yields

$$\kappa \propto D^{-\alpha_K}, \quad \gamma \propto D^0, \quad \text{and} \quad L(D) \propto D^{-\alpha_K},$$

$$(S47)$$

as promised. Here we have dropped sub-leading terms at large $D$. Scaling behavior for parameter scaling $L(P)$ follow via the dataset parameter duality.

### F.2 DUALITY BEYOND ASYMPTOTICS

Expressions (S37) and (S38) are related by changing projections onto finite feature set, and finite dataset even without taking any asymptotic limits. We thus expect the dependence of test loss on parameter count and dataset size to be related quite generally in linear feature models. See Section G for further details.

## G LEARNED FEATURES

In this section, we consider linear models with features coming from pretrained neural networks. Such features are useful for transfer learning applications (e.g. Kornblith et al. (2019); Kolesnikov et al. (2019)). In Figures S6 and S7, we take pretrained embedding features from an EfficientNet-B5

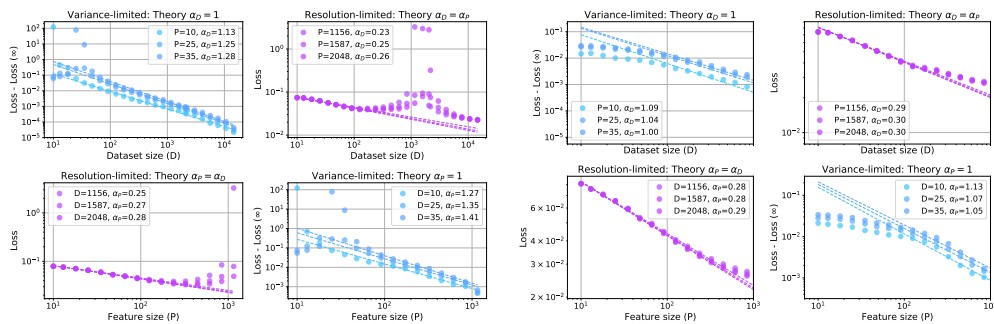

Figure S7: **Four scaling regimes exhibited by pretrained embedding features** Using pretrained embedding features of EfficientNet-B5 (Tan & Le, 2019) for fixed low regularization (**left**) and tuned regularization (**right**), we can identify four regimes of scaling using real CIFAR-10 labels.

model (Tan & Le, 2019) using TF hub[4]. The EfficientNet model is pretrained using the ImageNet dataset with input image size of $(456, 456)$. To extract features for the $(32, 32)$ CIFAR-10 images, we use *bilinear* resizing. We then train a linear classifier on top of the penultimate pretrained features. To explore the effect feature size, $P$, and dataset size $D$, we randomly subset the feature dimension and training dataset size and average over $5$ random seeds. Prediction on test points are obtained as a kernel ridge regression problem with linear kernel. We note that the regularization ridge parameter can be mapped to an inverse early-stopping time (Ali et al., 2019; Lee et al., 2020) of a corresponding ridgeless model trained via gradient descent. Inference with low regularization parameter denotes training for long time while tuned regularization parameter is equivalent to optimal early stopping.

In Figure S7 we see evidence of all four scaling regimes for low regularization (left four) and optimal regularization (right four). We speculate that the deviation from the predicted variance-limited exponent $\alpha_P = \alpha_D = 1$ for the case of fixed low regularization (late time) is possibly due to the double descent resonance at $D = P$ which interferes with the power law fit.

In Figure S6, we observe the duality between dataset size $D$ (solid) and feature size $P$ (dashed) – the loss as a function of the number of features is identical to the loss as function of dataset size for both the optimal loss (tuned regularization) or late time loss (low regularization).

In Figure S8, we also compare properties of random features (using the infinite-width limit) and learned features from trained WRN 28-10 models. We note that teacher-student models, where the feature class matches the target function and ordinary, fully trained models on real data (Figure 1a), have significantly larger exponents than models with fixed features and realistic targets.

The measured $\bar{\omega}_i$ – the coefficient of the task labels under the $i$-th feature (S45) are approximately constant as function of index $i$ for all teacher-student settings. However for real targets, $\bar{\omega}_i$ are only constant for the well-performing Myrtle-10 and WRN trained features (last two columns).

[4]https://www.tensorflow.org/hub

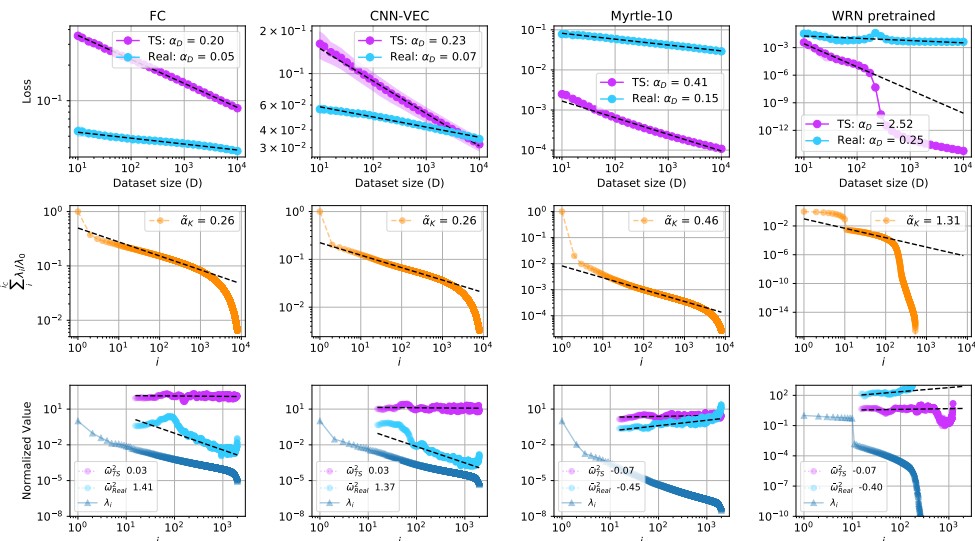

Figure S8: **Loss on the teacher targets scale better than real targets for both untrained and trained features** The first three columns are infinite width kernels while the last column is a kernel built out of features from the penultimate layer of pretrained WRN 28-10 models on CIFAR-10. The first row is the loss as a function of dataset size $D$ for teacher-student targets vs real targets. The observed dataset scaling exponent is denoted in the legend. The second row is the normalized partial sum of kernel eigenvalues. The partial sum's scaling exponent is measured to capture the effect of the finite dataset size when empirical $\alpha_K$ is close to zero. The third row shows $\bar{\omega}_i$ for teacher-student and real target compared against the kernel eigenvalue decay. We see the teacher-student $\bar{\omega}_i$ are approximately constant.

