# OpenReview forum: "Explaining Scaling Laws of Neural Network Generalization"
_ICLR.cc/2022/Conference — ICLR 2022 Submitted_

### Official Review · Reviewer_78Lv · 2021-10-26

**Correctness:** 3
**Technical Novelty And Significance:** 3
**Empirical Novelty And Significance:** 4
**Recommendation:** 5
**Confidence:** 2

**Main Review:**

The paper proposes a phenomenon called scaling laws of test loss for neural networks. In a nutshell, the test loss decreases by following a power law curve (exponent) with respect to width or data set size, while the other being fixed.  Specifically, the variance limited scaling follows simply from the existence of a well-behaved infinite data or infinite width limit, while the resolution-limited regime can be explained by
positing that models are effectively resolving a smooth data manifold. The proofs are for linear models (random feature) and pretrained models. Authors also observe several empirical relationships between datasets and scaling exponents: super-classing image tasks does not change exponents, while changing
input distribution (via changing datasets or adding noise) has a strong effect.

Overall, the paper studies a very interesting phenomenon. It's quite interesting to see such power law with respect to width and data set sizes with the other being fixed. Such good combination of interesting phenomenon with theoretical explanation is a great fit for ICLR. Hence, an accept has been recommended.

However, there are several concerns remaining.

1. It's counter-intuitive to see the accuracy follows power law for width. In the limit, it becomes the kernel regime, and we all know NTK is different from neural network. In this case, there must be finite width which perform better than infinite width? Then this is a counter-example to your phenomenon?

2. What's the practical different between variance limited and resolution limited? These two terms are bit confusing.

3. The data set size is reasonbale, but does not consider if we add more rare examples? If rare examples are of measure 0, then it's not reflected in your theory, but can make a big difference in practice?

4. In some cases, I find smaller network perform better (say width 32), larger width cannot work. How do you explain this?

**Summary Of The Paper:**

The paper proposes a phenomenon called scaling laws of test loss for neural networks. In a nutshell, the test loss decreases by following a power law curve (exponent) with respect to width or data set size, while the other being fixed.  Specifically, the variance limited scaling follows simply from the existence of a well-behaved infinite data or infinite width limit, while the resolution-limited regime can be explained by
positing that models are effectively resolving a smooth data manifold. The proofs are for linear models (random feature) and pretrained models. Authors also observe several empirical relationships between datasets and scaling exponents: super-classing image tasks does not change exponents, while changing
input distribution (via changing datasets or adding noise) has a strong effect.

**Summary Of The Review:**

Overall, the paper studies a very interesting phenomenon. It's quite interesting to see such power law with respect to width and data set sizes with the other being fixed. Such good combination of interesting phenomenon with theoretical explanation is a great fit for ICLR. Hence, an accept has been recommended.  However, authors need to clarify a few concerns.

---

> ### Author Response · Authors · 2021-11-19
> **Reply to reviewer 78Lv**
>
> We thank the reviewer for taking the time to read our work, for comments, and for the generally positive response. Please see a point by point response below.
>
> > The proofs are for linear models (random feature) and pretrained models.
>
> We prove results for these cases, as well as for any non-linear model satisfying the stated smoothness assumptions of Theorems 1, 2 and 3.
>
> > It's counter-intuitive to see the accuracy follows power law for width. In the limit, it becomes the kernel regime, and we all know NTK is different from neural network. In this case, there must be finite width which perform better than infinite width? Then this is a counter-example to your phenomenon?
> > In some cases, I find smaller network perform better (say width 32), larger width cannot work. How do you explain this?
>
> The fact that finite width networks can outperform wider networks is not a counterexample to our description. Indeed, there have been numerous studies where approaching the infinite width limit can be shown to be monotonic and exhibit predictable scaling in the NTK limit with infinite width networks performing either better or worse than finite width networks. In either case the difference scales as O(1/n) (Lee et al., NeurIPS 2019, Dyer and Gur-Ari, ICLR 2020, Andreassen‎ and Dyer 2020).
>
> There certainly are cases, e.g. near double descent, where monotonicity with width scaling may be broken depending on the training conditions. While understanding the effect of various training techniques used in deep learning on scaling laws is an important future direction, we believe having a controlled setting where the origin of scaling is easily identifiable, such as approaching the NTK limit, is a great starting point.
>
> As an aside, we’d like to correct that while some experiments do show classification error (1-accuracy) having power-law scaling, in this work we focus on power-law scaling in the loss.
>
> > What's the practical different between variance limited and resolution limited? These two terms are bit confusing.
>
> If X (P or D) >> Y (D or P), we define “variance-limited scaling” as the scaling of the loss with respect to X and “resolution-limited scaling” as the scaling of the loss with respect to Y. We hope this definition is clear from the text and if not we welcome any suggestions on how to clarify the presentation!
>
> Practically, the “variance limited” setting corresponds to scaling up either dataset size or number of parameters when these are already sufficiently large, with scaling coming from the reduction in variance around the limiting value. In contrast, the “resolution limited” setting corresponds to providing the limiting resource – i.e. scaling up the capacity when you are capacity limited (P << D) or scaling up the data when you are data limited (D << P). In our geometric picture, this amounts to providing increasingly more resolution when approximating a smooth function.  In practice, scaling up model sizes for overparameterized image classification is in the “variance-limited” setting whereas scaling up data corresponds to the “resolution-limited” setting.  On the other hand, in data rich settings (such as language modeling), model-size scaling is in the “resolution-limited” regime. Practically, we expect absolute performance to improve more dramatically as a result of the “resolution-limited” scaling, since this corresponds to adding the resource (either capacity or data) that is bottlenecking performance.
>
> > if we add more rare examples? ... then it's not reflected in your theory, but can make a big difference in practice?
>
> Indeed our theoretical setup assumes that train and test data are drawn IID from the same distribution. This means that we are only considering the case where rare examples can only appear proportionally to their probability in both train and test data. We do note that our experiments use the standard train and test splits for standard datasets, and thus the results appear to hold for the empirical distribution of examples present in those datasets.

---

### Official Review · Reviewer_TzFb · 2021-10-31

**Correctness:** 2
**Technical Novelty And Significance:** 3
**Empirical Novelty And Significance:** 3
**Recommendation:** 3
**Confidence:** 4

**Main Review:**

The decomposition of Variance-Limited Regime and Resolution-Limited Regime is somehow interesting. However, the reviewer find out it's another kind of bias-variance decomposition. The variance limited regime happens when the model size is constrained, if the rademencher complexity is bounded, then you have the 1/sqrt(data number) convergence, it's the variance in the statistics setting. The  Resolution-Limited Regime in the paper is the data is infinite and then the rate is the approximation rate.


For the Variance-Limited Regime, the statement is not clear. As I assume, the paper assumes the neural network lies in the  neural tangent kernel regime. Otherwise, the network weights are correlated, you can't do any concentration on width.  For the dataset scaling, the author claims "For every value of the weights, the training loss, thought of as a random variable over draws of a training set of size D, concentrates around the population loss, with a variance which scales as O(1/D)". This is wrong and is taught in statistical learning 101 course. The learned network is not independent with the data, so you can't use the concentration inequality, that's why the empirical process technique is introduced.(i.e. a uniform bound of all possible neural network is needed here (foundations of machine learning, Theorem 2.13 and chapter 3) .) Also the rate is not O(1/D) but is O(1/sqrt(D)) due to the  central limit theorem/Chernoff bound.

For the Resolution-Limited Regime, the 1/d scaling law is actually not surprising. The 1/d scaling law is actually the same as the non-parametric statistics [6]  and there is a line of research using a neural network to achieve this rate[1-5]. The bound the paper achieve is the same as the approximation rate obtained by [1-4]. They achieved N^{-s/d} approximation error for the s-holder function, and is standard and well-known in the literature. For the kernel setting, the nonparametric rate is from the Eigen decay  is also well known, see references in [5].

In section 2.3.1,the L(D,P)-L(D) should be O(1/sqrt{P}) but not O(1/P) due to central limit theorem/Chernoff bound,  similar errors appear many times in the paper as I think. In the case of linear models **with l2 loss**, I agree the variance will become O(1/P), but it's not the case for general model.

The paper is somehow interesting, however, the reviewer discovered that the author has little knowledge in basic statistics, as for example bias-variance decomposition, parametric rate (1/sqrt{N} variance fluctuation), and non-parametric rate (1/d scaling law). The notation in the paper is not well-defined and most of the "theorems" don't state the full set of assumptions of all theoretical results and even don't state the rigorous setting of the theorems. This doesn't seem like a rigorous paper but like a blog post. (For an example, in section 2.1 the expectation is an expectation on a different variable but the author use the same notation.)


[1] Schmidt-Hieber J. Nonparametric regression using deep neural networks with ReLU activation function[J]. The Annals of Statistics, 2020, 48(4): 1875-1897.

[2] Farrell M H, Liang T, Misra S. Deep neural networks for estimation and inference[J]. Econometrica, 2021, 89(1): 181-213.

[3] Suzuki T, Nitanda A. Deep learning is adaptive to intrinsic dimensionality of model smoothness in anisotropic Besov space[J]. arXiv preprint arXiv:1910.12799, 2019.

[4] Chen M, Jiang H, Liao W, et al. Efficient approximation of deep relu networks for functions on low dimensional manifolds[J]. Advances in neural information processing systems, 2019, 32: 8174-8184.

[5] Nitanda A, Suzuki T. Optimal rates for averaged stochastic gradient descent under neural tangent kernel regime[J]. arXiv preprint arXiv:2006.12297, 2020.

[6] Tsybakov A B. Introduction to Nonparametric Estimation[J]. Springer series in statistics, 2009: I-XII,1-214.

**Summary Of The Paper:**

This paper considered Variance-Limited Regime and Resolution-Limited Regime to explain the explored neural scaling law. In the Variance-Limited Regim, one fixed one of the D and P and in the Resolution-Limited Regime, one of the D and P is effectively infinite. The paper aims to provide a parametric rate scaling law in the Variance-Limited Regime  and a non-parametric scaling law in the Resolution-Limited Regime.

**Summary Of The Review:**

The paper is not well-written and theoretical results are not well-stated. The paper also igonres standard results in the statistical results, bias-variance decomposition vs the four regimes and the non-parametric rates. The reviewer also find out that the author lack of knowledge in basic statistical machine learning theory. Reviewer suggests the author write down the model considered (random feature, neural tangent kernel regime NN?) rigorously and state all the notation and theorem in detail.
The empirical experiment in the paper is interesting and meets the bar of a top conference, however the "theory" part is confusing, not novel and even something can be wrong.

---

> ### Author Response · Authors · 2021-11-20
> **Reply (2 out of 2) to reviewer TzFb**
>
> > … Also the rate is not O(1/D) but is O(1/sqrt(D)) due to the central limit theorem/Chernoff bound. …
>
> The O(1/x) vrs O(1/\sqrt{x}) bound seems to have been a central confusion of the reviewer.
>
> As a simple illustrative example, consider a random variable, a, with mean \bar{a} and variance and higher moments O(1/x). If we consider a polynomial function F(a). Than E[F(a)-F(\bar{a})] = O(1/x). We extend this basic example to more general smooth functions and through the training dynamics in the text and supplementary section D.
>
> The key difference between the above result, and the O(1/\sqrt{x}) Chernoff/Hoeffdings/McDiarmid's/Rademacher bounds the reviewer is referring to is whether we are bounding the difference, E[F(a)-F(\bar{a})], or the absolute value of the difference E[|F(a)-F(\bar{a})|]. For the latter, it is true that in general the strongest bound we can prove is O(1/\sqrt{x}).
>
> As an additional note, even if we consider the absolute value, if we further assume the trained network approaches a minimum of the loss then we can recover the O(1/x) scaling. This is discussed in the paragraph following equation S4 in the proof of Theorem 1.
> That being said, counterexamples to the O(1/D) scaling, either for non-smooth loss, or for diverging training time are presented in supplementary section D.3
>
> > In section 2.3.1,the L(D,P)-L(D) should be O(1/sqrt{P}) but not O(1/P) due to central limit theorem/Chernoff bound,
>
> This is incorrect and makes the same mistake as above. For additional examples of the O(1/P) scaling of linear models, see e.g. [Mei and Montanari 2019, Adlam and Pennington 2020a,b].
>
> >  In the case of linear models with l2 loss, I agree the variance will become O(1/P), but it's not the case for general model.
>
> Indeed, we do not claim this for general models. For wide neural networks (trained in the NTK limit) the variance, and thus the variance-limited loss difference scales as O(1/\srt{P}) = O(1/n) for any sufficiently smooth loss. This behavior is also seen empirically for the wide resnets studied. More generally if a trained network output approaches a limiting form for large P and if we can bound the fluctuations around this behavior, then we can bound the behavior of sufficiently smooth loss functions, we cite work making progress in this direction for the joint large width and depth limit [Hanin and Nica], but make no claims.
>
> > The 1/d scaling law is actually the same as the non-parametric statistics [6] and there is a line of research using a neural network to achieve this rate[1-5].
>
> We do not claim to be the first to derive a scaling exponent proportional to 1/d and indeed, we cite earlier works that have done so in both ML and the more general approximation theory setting. As we discuss also in our response to Reviewer f3X7, each of the references [1-5] study settings that are more structured and specific than ours, in the majority of cases with a focus on only one of the four scaling regimes we examine. We will be glad to add these references as citations, which we were unaware of. One of our contributions is to present a unified framework for when variance-limited and resolution-limited regimes occur, under a set of simple but quite general assumptions that are relevant for realistic settings of interest (e.g. such as SGD training of deep neural networks with cross-entropy loss) and in particular, emphasize that the non-parametric style scaling is associated with the resolution limited, but not variance limited regime. Furthermore, we show empirically that these scalings are actually observed in practice, rather than simply unachieved bounds.

---

> > ### Comment · Reviewer_TzFb · 2021-11-25
> > **Thank you for your response.**
> >
> > I'll not change my points due to the following points.
> > 1. You claimed "Our statement is correct and is not discussing trained weights but a fixed value one". Then the theory is not interesting at all. At the same time, as I said in the previous review, these assumptions should be listed explicitly in the manuscript.
> >
> > 2. You claim " references [1-5] study settings that are more structured and specific than ours" it's not fair. all the models you analysis is in the not-trained (at least no feature learning, linear kernel model) network. It' unfair to claim this
> >
> > 3. Regards O(1/sqrt{P}) and O(1/P), I agree it's O(1/P) for square models and is the case of the variance term reference [1-5].  But in the paper the loss function is k-th polynomial for k>=2. For k>2, the rate can even become faster. (You can refer to "Fast rate generalization bound" in the literature)
> >
> > This paper's finding is interesting, but if it's accepted, I think missing all the essential assumptions and missing discussion of the exist literature in the theory will mislead many readers.
> >
> > I don't see any improvement of the modified version of the paper in the case iclr can revise the draft. I would like to see the paper to list all the assumptions of the theorem, the network is untrained? kernel regime? define every notation at least in the appendix. Discuss every literature. For this paper actually mostly based on other paper's theory, I would like to suggest the author even not to list it as a theorem, just say it's supported by xxx.
> >
> > Even the paper delete all the "theory" just present the experiment, I can agree this paper to be accepted for the interesting experiments.

---

> ### Author Response · Authors · 2021-11-20
> **Reply (1 out of 2) to reviewer TzFb**
>
> We appreciate the reviewer taking the time to read our work and acknowledge the comments. We hope we have answered questions and corrected the misunderstandings the reviewer had below. We would like to note that the tone of the review, e.g “The reviewer also find out that the author lack of knowledge in basic statistical machine learning theory” is inappropriate for an ICLR review.
>
> > The variance limited regime happens when the model size is constrained, if the rademencher complexity is bounded, then you have the 1/sqrt(data number) convergence.
>
> This is not quite correct. The variance limited regime describes two limits, the limit mentioned, when model size is constrained and training dataset size is large and the limit of large model when the dataset size is constrained (for the latter we focus on overparameterized linear models and large width models). The convergence is 1/D (without the sqrt) in the former case (see below for additional discussion).
>
> > The Resolution-Limited Regime in the paper is the data is infinite and then the rate is the approximation rate.
>
> Again here, the resolution-limited regime describes two settings, in one case the data is infinite and this is the approximation rate as a function of model size (as mentioned). In the other case, the model capacity is infinite, and this is the approximation rate with respect to training dataset size.
>
> > For the Variance-Limited Regime, the statement is not clear. As I assume, the paper assumes the neural network lies in the neural tangent kernel regime.
>
> For the variance-limited regime with respect to parameters, we indeed assume that, in the width goes to infinity limit, we enter the NTK regime. Though as mentioned this is not guaranteed [Lee et al. 2019, Lewkowycz et al. 2020, Huang et al. 2020], it does describe the large width properties for many networks and optimization procedures [Jacot et al. 2018, Lee et al. 2019, Chizat et al., 2019]. In section 2.3 we also consider linear models directly. For the variance-limited regime with respect to dataset size, there is no such assumption.
>
> > the author claims " For any value of the weights, the training loss, thought of as a
> random variable over draws of a training set of size D, concentrates around the population loss, with a variance which scales as O(1/D)". This is wrong … The learned network is not independent with the data,
>
> Our statement is correct and is not discussing trained weights. For any fixed value (ie fixed independent of the draw of training data) of the weights, the result follows from the fact that the loss with fixed weights is a sample estimate over D iid draws. We are happy to clarify that this sentence refers to fixed, data independent values of the weights.
>
> Of course, the reviewer is correct that what we ultimately care about is the value of the loss after training, which introduces correlation between the weights and data. That is why the text in that section continues to discuss the optimization procedure as a map from initial weights and training loss to our trained weights.
>
> The discussion presented in section 2.1 is informal and intended to provide intuition. In section D, we prove this statement when training for fixed time and with polynomial loss (for GD and SGD). However we expect it to hold more generally, and the experiments in Figure 1 top left are performed, for instance, with cross entropy loss.

---

### Official Review · Reviewer_f3X7 · 2021-11-02

**Correctness:** 3
**Technical Novelty And Significance:** 2
**Empirical Novelty And Significance:** 3
**Recommendation:** 3
**Confidence:** 3

**Main Review:**

Neural scaling laws have become a popular topic in machine learning, so the theoretical understanding, which this submission attempts to address, is an important research problem. However, I have the following concerns:

**(i)** Many important prior works (related to the results presented in this submission) were not cited. The authors should discuss the difference from and improvement upon these papers. In particular,
1. If we do not take optimization into account, then the so-called "scaling law" of neural network has already been extensively studied in the nonparametric literature. For example, [Schmidt-Hieber] and [Suzuki] derived the learning rates of deep neural network in fitting certain target functions; such analysis is usually divided into two parts, the estimation error, which scales like $D^{-\alpha}$, and the approximation error, which scales with $P^{-\beta}$, where the exponents $\alpha, \beta$ depend on the problem dimensionality and smoothness of the target function.
2. If we restrict ourselves to linear/kernel models, then power-law generalization error rates are fairly standard and well-known results (e.g., see [Caponnetto and De Vito]). Moreover, for neural networks in the kernel regime, the role of optimization (i.e., the number of training steps) can also be incorporated in the analysis, as in [Nitanda and Suzuki].
3. For the random features model, the "dual" relation between the model width and training set size has been rigorously studied in [Ghorbani et al.] and [Mei et al.]. Note that [Ghorbani et al.] has been publicly available for more than 2 years, so I don't think the current submission can be treated as concurrent work.

- Caponnetto and De Vito, FOCM 2007. https://link.springer.com/article/10.1007/s10208-006-0196-8.
- Schmidt-Hieber, Annals of Statistics 2020. https://arxiv.org/pdf/1708.06633.pdf.
- Suzuki, ICLR 2018. https://arxiv.org/pdf/1810.08033.pdf.
- Ghorbani et al., Annals of Statistics 2021.  https://arxiv.org/pdf/1904.12191.pdf.
- Suzuki and Nitanda, ICLR 2021. https://arxiv.org/pdf/2006.12297.pdf.
- Mei et al., 2021. https://arxiv.org/pdf/2101.10588.pdf.

**(ii)** The theoretical results are somewhat underwhelming, and some discussions lack rigor.
1. It is not clear whether the $O(1/D)$-fluctuation is relevant in practical neural network training, because it omits other dependencies. For example, if we naively couple the finite-sample (stochastic) gradient descent with the population counterpart, then without additional assumptions, the difference between the two trajectories can blow up exponentially in time. Hence this bound would be uninformative beyond the early phase of training. Moreover, this does not provide any insight on the scaling law of training speed.
2. The smoothness parameters in Theorem 2 and 3 are treated as constants independent to $D$ or $P$. Why wouldn't the Lipschitz constant of the learned interpolator depend on $D$, especially in the noisy setting?
3. Many findings in Section 2.3 and 2.4 have little to do with actual neural network. For example, while a wide model can be "frozen" in the kernel regime under certain initialization, it is not clear why a trained network in the $D\gg P$ regime would resemble a random features model with similar spectral properties.
4. For the replica computation, the authors should elaborate on the difference from the following papers, both of which derived power-law decay of generalization error for kernel regression.
- Spigler et al., J. Stat. Mech. 2020. https://arxiv.org/pdf/1905.10843.pdf.
- Bordelon et al., ICML 2020. https://arxiv.org/pdf/2002.02561.pdf.

**Additional Comments**

- For Section 2.3.2, the authors should comment on when one can expect such power-law decay of the kernel spectra to hold true. Intuitively this may not happen when $D\asymp d$ (for some notion of intrinsic dimensionality $d$) -- in this regime the kernel eigenvalues are usually bounded away from 0.

**Summary Of The Paper:**

This paper characterized the scaling of generalization error with respect to the number of training samples $D$ and parameters $P$ for certain smooth estimators in two regimes: the variance-limited regime and the resolution-limited regime; this divide is somewhat analogous to the parametric vs. nonparametric estimation. The authors also argued that the large-width and large-sample limit have similar scaling behavior. The findings are supported by empirical evidence.

**Summary Of The Review:**

I evaluated this submission primarily as a theory paper; in my opinion the current theoretical contribution is below the acceptance bar, for reasons listed above. I am happy to adjust my score if the authors can address my concerns.

---

> ### Author Response · Authors · 2021-11-20
> **Reply (3 out of 3) to reviewer f3X7**
>
> > ii 3.)  … little to do with actual neural network. it is not clear why a trained network in the D≫P regime would resemble a random features model with similar spectral properties.
>
> None of the results presented claim that general neural networks are described by a linear random feature model. For general models, we rather i) present arguments relying on smoothness assumptions ii) Present empirical evidence of power law scaling behavior with qualitatively different characteristics depending on whether we are in the resolution or variance limited regime.
>
> The purpose of section 2.3 is to show that even models as simple as linear random feature models (provably) exhibit the four regimes we observe in more realistic settings as long as one includes assumptions about the data.
>
> That being said, there are various intuitions directly connecting the linear model picture with realistic networks. It has been suggested that near the end of training, or during finetuning last layer weights move the most. Alternatively, sometimes finetuning is done by training a single linear readout over pretrained weights. In the former case approximately and the latter case directly, the final stage of training can be related to a linear model on pretrained features. The analysis in section 2.3 holds equally well when the random features are random subsets of pretrained features. If we imagine that changing network size is similar to randomly drawing from a large pool of pretrained features, than this gives a direct map between the late stage of training / finetuning and the linear model studied in 2.3. The empirical behavior of linear models with pretrained features is shown in Figure S7.
>
> > ii 4.) The authors should elaborate on the difference from Spigler et al., and Bordelon et al.,
>
> Both papers only study dataset resolution-limited scaling, which is only one out of the four regimes we investigate. Spigler et al. is restricted to Gaussian and Laplace kernels, making use of their translation-invariance. We do not have a restriction on the choice of kernel / random features. Furthermore, their teacher model is a Gaussian process. Spigler et al contains some discussion on the effective dimensionality and the nearest-neighbor distance scaling, but this is evaluated in a rather restricted setting (e.g. nearest-neighbor distance in input space, for data that lives on the sphere).
>
> We do make use of the replica theory derivations from Bordelon et al in analysis of resolution-limited scaling for the teacher-student random features model. In later versions of their work, the authors utilize the power-law behavior of the kernel eigenvalue spectrum in their formulas, leading to power-law resolution-limited dataset scaling. However, as we noted above, this work only captures one of the four scaling regimes we study, and even in this single regime does not cover the more general case of neural networks, which we also investigate.
>
> > For Section 2.3.2, the authors should comment on when one can expect such power-law decay of the kernel spectra to hold true.
>
> In section 2.3.2 Data Manifolds and Kernels, we attempted to present when one can expect such power-law decay. In particular, smooth kernels on d-dimensional manifolds have spectra satisfying a power law bound related to smoothness and manifold dimension. Our expectation is that generic smooth kernels without additional symmetry will saturate these bounds, though we do not have a precise set of criteria for a kernel to saturate. We would be happy to clarify this further if the reviewer has any specific suggestions.

---

> ### Author Response · Authors · 2021-11-20
> **Reply (2 out of 3) to reviewer f3X7**
>
> > i 3.) For the random features model, the "dual" relation between the model width and training set size has been rigorously studied in [Ghorbani et al.] and [Mei et al.].
>
> First, there is a crucial difference in the theoretical setup with these works. Both [Mei et al.] and [Ghorbani et al.] study the triple limit $n, N, d \rightarrow \infty$ (dataset size, number of random features, and input dimensionality, respectively.) We do not study this regime. Instead, for us the input dimensionality d is finite, while n, N are large.
>
> Nonetheless, indeed [Mei et al.] find the “dual” relation, which we were not aware of, and we will be happy to cite in the revision.
>
> We do not find discussion on the “dual” relation in [Ghorbani et al.]. Moreover, [Ghorbani et al.] treat the setting of one-hidden layer neural networks in the kernel regime, with input data restricted to the d-dimensional sphere. We do not restrict the setting giving rise to our features or the input space – except through the assumption on the kernel spectrum.
>
> > ii 1.) It is not clear whether the O(1/D)-fluctuation is relevant in practical neural network training, … for example, if we naively couple the finite-sample (stochastic) gradient descent … the difference between the two trajectories can blow up exponentially in time.
>
> This is a good question. Empirically, in Figure 1 - top left - we provide evidence that the O(1/D) scaling can be seen in realistic neural networks, including networks trained via SGD. In supplementary section D we give intuition for why this scaling holds even for models trained via SGD – for times fixed as D is taken large, and for models converging to local minima. We  prove this for the case of polynomial loss and fixed time, but expect it to hold more generally (and indeed the experimental setup of Figure 1 top-left is CE loss). As noted, there are certainly cases where this does not hold, including cases with non-smooth loss, or where one does not fix the training time and the network function has some divergence as T is taken large. We presented such counterexamples in section D.3
>
> > ii 2.) The smoothness parameters in Theorem 2 and 3 are treated as constants independent to D or P. Why wouldn't the Lipschitz constant of the learned interpolator depend on D, especially in the noisy setting?
>
> Theorems 2 and 3 are correct as written (the Lipschitz constant is constant by assumption, of course), but this in general could lead to loose upper bounds. We can derive a similar bound where we explicitly see why there is a “constant” coefficient originating from various Lipschitz constants – this is done in Section 2.2 - "From Bounds to Estimates." As we write in that section, the test error per point can be expanded as:
>
> $  L( x_{test} ) = \sum_{m=n \geq 2}  a_{m}( x_{train}) ( x_{test} - x_{train} )^{m} $
>
> The collection of coefficients $ \{ a_{m}(x_{train}) \} $ in principle have a dependence on dataset size D as well (or model size P for P-scaling). However, assuming a smooth large D (or P) limit, they too can be expanded about their infinite D limiting value (or likewise, infinite P for P-scaling). For instance, $a_2(x_{train}; D) = a_{2}(x_{train}; D \rightarrow \infty) + \text{finite-D corrections}$. These finite-D corrections are subleading resulting in:
>
> $ L(x_{test}) = a_{2}(x_{train}; D \rightarrow \infty) D^{-2/d} + \text{ corrections} $
>
> Therefore, asymptotically, the dominant D-scaling is still expected to be power law with exponent 2/d generically (or 4/d for piecewise linear networks, such as Relu neural networks) and a constant coefficient $a_{2}$ (playing the role of the Lipschitz constant earlier).

---

> ### Author Response · Authors · 2021-11-20
> **Reply (1 out of 3) to reviewer f3X7**
>
> Thank you for taking the time to read our paper, for your detailed feedback and the suggested references. Please see our point-by-point responses below.
>
> > I evaluated this submission primarily as a theory paper
>
> We would like to point out that our paper contributes a mixture of theory and empirical results, and our goal was to determine scaling that would govern realistic settings. Hence, we think that evaluating our paper purely on theoretical grounds misses sizable contributions that tie theory and experiments together. Nonetheless, we address all of the concerns with respect to theoretical results point by point below.
>
> One goal of our work was to identify a fairly simple, general set of assumptions that would be sufficient to give rise to the four regimes (2 in each of the variance and resolution-limited scalings) as well as to disentangle which scaling behavior should be expected in which asymptotic regime. Comparison with the cited papers below reveals that their analysis was often done in more structured settings. One contribution of our work is to put forth a unified understanding and identify a simple set of assumptions that gives rise to all four scaling regimes and exhibit that the consequences of these general assumptions appear to hold empirically in a variety of realistic training regimes, models and datasets.
>
> i ) Thank you for pointing us to these papers. We were not aware of [Caponnetto and De Vito], [Schmidt-Heber], [Suzuki], [Nitanda and Suzuki], and [Mei, et al.] and will be happy to update the paper with citations to all references you have listed. Nonetheless, there are differences with each of these past works, which we discuss case-by-case.
>
> > i 1.) If we do not take optimization into account, then the so-called "scaling law" of neural network has already been extensively studied in the nonparametric literature.
>
> Again, we emphasize that one of our main contributions is to (i) identify simple, general assumptions that give rise to all four scaling regimes and (ii) demonstrate that realistic networks realize these scalings.
>
> The works you mention are in a more structured, specific setting than ours. [Schmidt-Heber] assumes sparsity, is restricted to Relu neural networks, and assumes a compositional target function . We do not have such restrictions. Furthermore, the exponent derived in their Eq. 8 is strictly < 1, so these results would not capture the variance-limited scaling regime we study. [Suzuki] also appears to be restricted to Relu networks, with exponents that are again < 1 and hence do not capture variance-limited scaling. The Besov space assumption on the target function is different from what we consider.
>
> > i 2.) If we restrict ourselves to linear/kernel models, then power-law generalization error rates are fairly standard and well-known results (e.g., see [Caponnetto and De Vito]).
>
> Thank you for this reference, which we were not aware of. We agree that power-law dataset scaling for kernel methods is known; indeed, we cite [Spigler, et al] and [Bordelon, et al] for these contributions in our work. We emphasize again that one of our main contributions is to identify a single set of fairly simple, general assumptions that give rise to all four scaling regimes. These other references mentioned study a particular structured setting, which then obscures that they are special cases of a more general classification of regimes.
>
> For instance, note that [Caponnetto and De Vito] only study the linear/kernel regime for dataset scaling. We additionally have results for model scaling. Their Theorem 1 is a different result than ours; in the teacher-student kernel setting, we find the dataset scaling exponent simply depends linearly on the power-law exponent of the eigenvalue spectrum. Furthermore, through our empirics we show that these scalings are actually realized in kernels on real data, rather than just upper bounds. We note that [Nitanda and Suzuki] treat stochastic gradient-descent with averaged parameters in the online setting. We study empirical risk with finite data, and so our dataset scaling is not directly comparable to theirs. Moreover, they do not give the explicit dependence on the model size.

---

> ### Comment · Reviewer_f3X7 · 2021-11-27
> **Reply to authors**
>
> Thanks for the detailed reply. It is unfortunate that the authors chose not to revise the paper during the rebuttal period. In my opinion the current submission requires a major revision to be accepted. I'm therefore inclined to keep my score.
> Due to the limited time, I will only comment on a few points from the authors' response.
>
> 1. *"We agree that power-law dataset scaling for kernel methods is known."*
>
> The authors should do a better job in discussing existing results. For example, the power law spectra was analyzed in version 4 of [Bordelon et al.], which appeared in June 2020. If the authors want to claim that derivation in this submission is concurrent, then this should be stated explicitly. Otherwise, the authors should either cite the prior result directly or explain the difference (it’s not enough to simply state *”both papers only study dataset resolution-limited scaling”*).
> Also, there are many well-known works on nonparametric rates in addition to [Caponnetto and De Vito], and in many cases these results match the precise rates derived for Gaussian design. See [Cui et al.] for details and appropriate references.
>
> Cui et al, NeurIPS 2021. https://arxiv.org/pdf/2105.15004.pdf.
>
> 2. *"We do not find discussion on the “dual” relation in [Ghorbani et al.]."*
>
> [Ghorbani et al.] showed the prediction error of random features model in the large-sample limit (as a function of model width) resembles the error in the large-width limit (as a function of sample size) -- this is analogous to the dual relation discussed in this submission. This result is then refined in [Mei et al.] to cover any polynomial scaling of the width and sample size.
>
> 3. *"our main contributions is to (i) identify simple, general assumptions that give rise to all four scaling regimes."*
>
> I'm not sure if I agree with this statement. The assumptions made in those prior works allow for rigorous analysis of the power law rates. Given the informal nature of the arguments presented in this submission, I don't think the generality of assumptions is directly comparable. For example, the Besov space assumption is a specific form of smoothness constraint -- similar assumptions (but less precise) have also been made this submission.
>
> 4. *"asymptotically, the dominant D-scaling is still expected to be power law with exponent 2/d generically"*
>
> I'm not sure if I follow this reasoning. I'd appreciate if the authors could outline the assumptions on the data and student model under which the Lipschitz constant of the interpolating predictor is asymptotically independent to the training set size.

---

### Author Response · Authors · 2021-11-20
**Thank you for your reviews**

We thank the reviewers for taking the time to read our work and for the feedback. Please see individual responses below.

---

### Decision · Program_Chairs · 2022-01-20

**Decision:**

Reject

**Comment:**

This paper investigates the scaling laws of neural networks with respect to the number of training samples $D$ and parameters $P$ for some estimators in two regimes: the variance-limited regime and the resolution-limited regime. The theoretical results are supported by some numerical experiments.

Unfortunately, the paper has several critical issues, in particular, in its novelty and technical correctness.
1. The theoretical analyses lack much of their rigor. The assumptions and problem setups are not precisely introduced. Accordingly, the statement of each theorem is presented in an inaccurate way. Moreover, some theoretical consequences contain technical flaws (e.g., $1/P$ should be replaced by $1/\sqrt{P}$ without an appropriate assumption on the loss function such as strong convexity and smoothness).
2. Many of the presented results are already known in the literatures. It is unfortunate that the authors did no cite relevant existing literatures and did not discuss its novelty compared with the existing work.

For those reasons, this paper lacks its novelty and the quality of the paper is not sufficient to be accepted.
I recommend the authors to thoroughly survey the literature of the statistical learning theory from classic nonparametric regression analyses to recent advances on overparameterization.